# Leveraging Google's Tensor Processing Units for tsunami-risk mitigation planning in the Pacific Northwest and beyond

Ian Madden[1], Simone Marras[2], and Jenny Suckale[1,3]

[1]Institute for Computational and Mathematical Engineering, Stanford University
[2]Department of Mechanical Engineering & Center for Applied Mathematics and Statistics, New Jersey Institute of Technology
[3]Department of Geophysics, Doerr School of Sustainability, Stanford University
**Correspondence:** Simone Marras (smarras@njit.edu)

**Abstract.** Tsunami-risk mitigation planning has particular importance for communities like those of the Pacific Northwest, where coastlines are extremely dynamic and a seismically-active subduction zone looms large. The challenge does not stop here for risk managers: mitigation options have multiplied since communities have realized the viability and benefits of nature-based solutions. To identify suitable mitigation options for their community, risk managers need the ability to rapidly evaluate several different options through fast and accessible tsunami models, but may lack high-performance computing infrastructure. The goal of this work is to leverage Google's Tensor Processing Unit (TPU), a high-performance hardware accessible via the Google Cloud framework, to enable the rapid evaluation of different tsunami-risk mitigation strategies available to all communities. We establish a starting point through a numerical solver of the nonlinear shallow-water equations that uses a fifth-order Weighted Essentially Non-Oscillatory method with the Lax-Friedrichs flux splitting, and a Total Variation Diminishing third-order Runge-Kutta method for time discretization. We verify numerical solutions through several analytical solutions and benchmarks, reproduce several findings about one particular tsunami-risk mitigation strategy, and model tsunami runup at Crescent City, California whose topography comes from a high-resolution Digital Elevation Model. The direct measurements of the simulations performance, energy usage, and ease of execution show that our code could be a first step towards a community-based, user-friendly virtual laboratory that can be run by a minimally trained user on the cloud thanks to the ease of use of the Google Cloud Platform.

# 1 Introduction

The coast of the Pacific Northwest, from Cape Mendocino in California to Northern Vancouver Island in Canada as depicted in Fig. 1, is located on the seismically active Cascadia subduction zone (Heaton and Hartzell, 1987; Petersen et al., 2002). Along the 1200-km-long Cascadia subduction zone, there have been no large, shallow subduction earthquakes over the approximately 200 years of modern-data monitoring, but large historic earthquakes have left an unambiguous imprint on the coastal stratigraphy

(Clague, 1997). Sudden land level change in tidal marshes and low-lying forests provide testimony of 12 earthquakes over the last 6700 years (Witter et al., 2003), including one megathrust event that ruptured the entirety of the current Cascadia subduction zone in 1700 BC (Nelson et al., 1995; Wang et al., 2013). The event created a massive tsunami that swept across the entire Pacific Ocean devastating communities as far away as Japan (Satake et al., 1996; Atwater et al., 2011). Current seismic-hazard models estimate

that the probability of another magnitude 9+ earthquake happening within the next 50 years is about 14% (Petersen et al., 2002).

A magnitude 9+ Cascadia earthquake and tsunami occurring during modern times would devastate many low-lying communities along the Pacific Northwest. A recent assessment suggests that deaths and injuries could exceed tens of thousands and entails economic damages in the order of several billions

of dollars for Washington and Oregon State (see, e.g., Knudson and Bettinardi, 2013; Gordon, 2012), with potentially severe repercussions for the entire Pacific coast and country as a whole. The tsunami itself would put tens of thousands at risk of inundation, and threaten the low-lying coastal communities specifically in the Pacific Northwest with very little warning time for evacuation (Gordon, 2012). But how to confront this risk? Traditionally, the most common approach to reducing tsunami risk is the

construction of sea walls, but this hardening of the shoreline comes at a staggering price in terms of the economic construction costs (e.g., 245 miles of sea walls in Japan cost $12.74 billion) and in terms of long-term negative impact on coastal ecosystems (Peterson and Lowe, 2009; Dugan and Hubbard, 2010; Bulleri and Chapman, 2010) and shoreline stability (Dean and Dalrymple, 2002; Komar, 1998).

A potentially appealing alternative to sea walls are so-called hybrid approaches. Hybrid risk miti-

gation combines nature-based elements and traditional engineering elements to reduce risk while also providing co-benefits to communities and ecosystems. An example of a hybrid approach to tsunami-risk mitigation is a coastal mitigation park: A landscape unit on the shoreline built specifically to protect

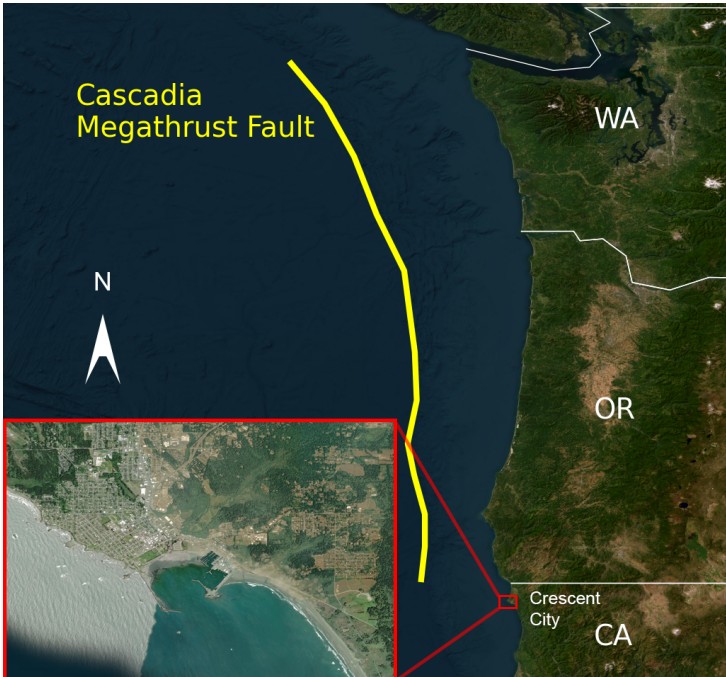

**Figure 1.** Map of the Cascadia Subduction Zone in the Pacific Northwest of the United States. Relative location of Crescent City with respect to the Megathrust Fault Line, with a more detailed picture of the Crescent City coastline. Esri provided access to the satellite imagery. Crescent City map at high resolution provided by Maxar. Pacific Northwest Map provided by Earthstar Geographics.

communities or critical infrastructure and provide vertical evacuation space, in the styles of Fig. 2. Com-
munities across the Pacific Northwest are increasingly considering these nature-based or hybrid options

(Freitag et al., 2011), but many important science questions regarding protective benefits and optimal
design remain open (Lunghino et al., 2020; Mukherjee et al., 2023). This gap is particularly concerning
given that existing models show that a careful design is necessary to avoid potential adverse effects
(Lunghino et al., 2020). The design of current mitigation parks, such as the one being built in Constitu-
ción, Chile, is not yet underpinned by an in-depth quantification of how different design choices affect

risk-reduction benefits, partly because numerical simulations of tsunami impacts are computationally
expensive.

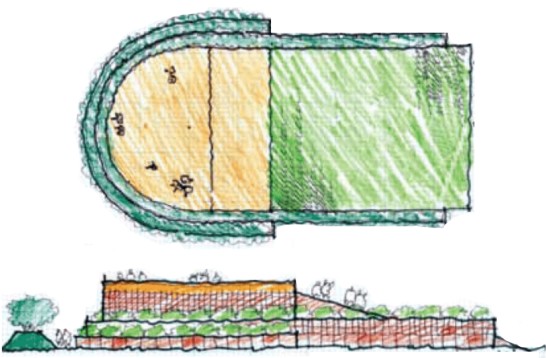

**Figure 2.** Map view (top) and side view (bottom) of a proposed tsunami-mitigation berm as designed by Project Safe Haven. The berm provides vertical evacuation space for the adjacent community and could also lower the onshore energy flux that drives the damage created by tsunami impact. We show this design as one example of a hybrid approach to tsunami-risk mitigation as it combines an engineered hill and ramp with natural vegetation. Sketches are adapted from Freitag et al. (2011).

The goal of this paper is to leverage Google's Tensor Process Units (TPUs) for enabling a fast evaluation of different mitigation park design and ultimately advancing evidence-based tsunami-risk mitigation planning rooted in quantitative assessments. TPUs are a new class of hardware accelerators developed by Google with the primary objective of accelerating machine learning computations. They are accessible via the Google Cloud Platform (Jouppi et al., 2017) and have recently been used for many different applications in computational physics and numerical analysis (Wang et al., 2022; Hu et al., 2022; Lu et al., 2020b, a; Belletti et al., 2020). We build upon and extend an existing implementation (Hu et al., 2022) to simulate the impact of idealized tsunamis on the coastline. Our implementation of the shallow water equations includes a non-linear advective term, not considered in (Hu et al., 2022), within the software framework based on Google's `TensorFlow` necessary to execute code on the TPU. We discretize the new equations using the Weighted Essentially Non-Ocillatory (WENO) method (Liu et al., 1994) and a third-order Runge-Kutta in time.

Numerical simulations of tsunamis have contributed to our understanding of and ability to mitigate wave impacts for many decades now, starting from the early work by Isozaki and Unoki (1964) for Tokyo Bay, and Ueno (1960) for the Chilean coast. The ability to capture the rupture mechanism that

generates the initial condition for tsunami propagation then enabled the reproduction of many historical tsunamis (Aida, 1969, 1974). Since then, many numerical models have been developed to simulate tsunami generation (Borrero et al., 2004; Pelties et al., 2012; López-Venegas et al., 2014; Galvez et al.,

2014; Ulrich et al., 2019), propagation (Titov et al., 2005; LeVeque et al., 2011; Chen et al., 2014; Allgeyer and Cummins, 2014; Abdolali and Kirby, 2017; Bonev et al., 2018; Abdolali et al., 2019), and inundation (Lynett, 2007; Park et al., 2013; Leschka and Oumeraci, 2014; Chen et al., 2014; Marsooli and Wu, 2014; Maza et al., 2015; Oishi et al., 2015; Prasetyo et al., 2019; Lunghino et al., 2020) by solving different variations of the shallow water and Navier-Stokes equations.

The list of existing numerical models is long and was recently reviewed by Marras and Mandli (2021) and Horrillo et al. (2015). Some commonly used ones are FUNWAVE (Kennedy et al., 2000; Shi et al., 2012), pCOULWAVE (Lynett et al., 2002; Kim and Lynett, 2011), Delft3D (Roelvink and Van Banning, 1995), GeoCLAW (Berger et al., 2011), NHWAVE (Ma et al., 2012), Tsunami-HySEA (Macías et al., 2017; Macías et al., 2020b, a), FVCOM (Chen et al., 2003, 2014). Our work here relies on well-

known numerical techniques to solve idealized tsunami problems. Its novelty lies in demonstrating the capability and efficiency of TPUs to solve the non-linear shallow water equations to model tsunamis.

We intentionally use a hardware infrastructure that is relatively easy to use and accessible without specific training in high-performance computing. For the TPU infrastructure that we use here, comprehensive tutorials using Google Colab are available at https://cloud.google.com/tpu/docs/colabs. The

TPU may increasingly become a standard hardware on which physics-based machine-learning algorithms will be built (Rasp et al., 2018; Mao et al., 2020; Wessels et al., 2020; Fauzi and Mizutani, 2020; Liu et al., 2021; Kamiya et al., 2022). Through its relative ease of access and potential for rapid simulation capabilities, cloud computing provides a valuable alternative to higher performance computing clusters (Behrens et al., 2022; Zhang et al., 2010), particularly for communities with limited access to

local clusters. By leveraging these benefits of the Google Cloud TPU, we propose our implementation as one step towards a community-based, user-friendly virtual laboratory that can be run by a minimally trained user on the cloud. The tool, which is freely available on Github at (tsunamiTPUlab, 2023) under an Apache License, Version 2.0 for collaborative open source software development, can be modified to include machine learning capabilities and, eventually, extended to coupled models for earthquake

generation, inundation, and human interaction.

## Methods

### Numerical approximation

We model tsunami propagation and runup with the 2D non-linear shallow water equations in the conservative formulation with a source term in a Cartesian coordinate system. Letting $\mathbf{x} = (x, y)$ denote position, we define $u(\mathbf{x}, t)$ and $v(\mathbf{x}, t)$ as the flow velocities in the $x$ and $y$ directions, respectively. We define $h(\mathbf{x}, t)$ as the dynamic water height and $b(\mathbf{x})$ as the imposed bathymetry, meaning that the quantity $h + b$ represents the water surface level. We solve for $h$, $hu$, and $hv$ in our implementation. We further place a lower bound $h \geq \epsilon$ in all cells, meaning no properly 'dry' cells are present, and handle cells with water depth $h = \epsilon$ (the value of $\epsilon$ is problem dependent and on the order of centimeters) using the TPU code provided by Hu et al. (2022). This ensures that no flux arrives from those cells with water height $\epsilon$. While this approach has some important drawbacks (Kärnä et al., 2011), it has been used extensively (Bates and Hervouet, 1999; Bunya et al., 2009; Gourgue et al., 2009; Nikolos and Delis, 2009; Marras et al., 2018) and is sufficient to keep the code stable within the scope of this study. This leads to the following system of equations, a set very similar to that suggested by Xing and Shu (2005)

$$\frac{\partial}{\partial t} h + \frac{\partial}{\partial x}(hu) + \frac{\partial}{\partial y}(hv) = 0 \tag{1}$$

$$\frac{\partial}{\partial t}(hu) + \frac{\partial}{\partial x}\left(\frac{(hu)^2}{h} + \frac{1}{2}g(h^2 - b^2)\right) + \frac{\partial}{\partial y}(huv) = -g(h+b)\frac{\partial b}{\partial x} - \frac{gn^2\sqrt{(hu)^2 + (hv)^2}}{h^{7/3}}(hu) \tag{2}$$

$$\frac{\partial}{\partial t}(hv) + \frac{\partial}{\partial x}(huv) + \frac{\partial}{\partial y}\left(\frac{(hv)^2}{h} + \frac{1}{2}g(h^2 - b^2)\right) = -g(h+b)\frac{\partial b}{\partial y} - \frac{gn^2\sqrt{(hu)^2 + (hv)^2}}{h^{7/3}}(hv), \tag{3}$$

where $g = 9.81$ ms$^{-2}$ is the acceleration of gravity, and $n$ is the Manning friction coefficient. Note that the left-hand-side of our formulation of the shallow water equations includes the full nonlinear advection terms.

For ease of future notation, we let $\mathbf{u} = \begin{bmatrix} h & hu & hv \end{bmatrix}^T$, and we rewrite the above equations in a vector form, namely:

$$\frac{\partial \mathbf{u}}{\partial t} + \frac{\partial \mathbf{F}}{\partial x} + \frac{\partial \mathbf{G}}{\partial y} = \mathbf{S} \tag{4}$$

where $\mathbf{F}$ and $\mathbf{G}$ are the fluxes in the $x$ and the $y$ directions for the vector $\mathbf{u}$, and $\mathbf{S}$ is a source term arising from variations in topography and Manning coefficient.

We implement these shallow-water equations using the finite volume method whereby the half-step flux and height values are determined through a fifth-order WENO scheme (Liu et al., 1994; Jiang

and Shu, 1996). We approximate solutions to cell-wise Riemann problems by formulating fluxes using the Lax-Friedrichs method as in LeVeque (2011). We formulate the bed source term as suggested by Xing and Shu (2005), and formulate the friction term explicitly rather than using the implicit process suggested by Xia and Liang (2018). We use a third-order, Total Variation Diminishing Runge-Kutta scheme (Shu, 1988) to step the numerical solution forward in time.

We begin the discretization of the equation in continuous variables $t$, $x$, and $y$, using respective step sizes $\Delta t$, $\Delta x$, and $\Delta y$, which indicate the distance between consecutive integral steps in the discrete indices $n$, $i$, and $j$, respectively. We use the fifth-order WENO scheme in the $x$ and $y$ directions, where two values of each quantity $h$, $hu$, and $hv$ are determined at each half-step of $x$ and $y$. These two values correspond to a positive and negative characteristic, due to the nature of the footprint that is chosen at a given point. In other words, given the conservative form with relevant variable $\mathbf{u}$, $\mathbf{u}_{i,j}$ centered on a finite volume cell, we label these outputs of WENO:

$$\mathbf{u}^{+}_{i+\frac{1}{2},j}, \mathbf{u}^{-}_{i+\frac{1}{2},j} \quad \text{for WENO in } x, \quad \text{or} \quad \mathbf{u}^{+}_{i,j+\frac{1}{2}}, \mathbf{u}^{-}_{i,j+\frac{1}{2}} \text{for WENO in } y. \tag{5}$$

From here, we use the Lax-Friedrichs method to approximate flux values that serve as solutions to the Riemann problem; i.e. we approximate

$$\mathbf{F}_{i+\frac{1}{2},j} = \frac{1}{2}\left[\mathbf{F}(\mathbf{u}^{+}_{i+\frac{1}{2},j}) + \mathbf{F}(\mathbf{u}^{-}_{i+\frac{1}{2},j}) - \alpha_{\mathbf{u}}\left(\mathbf{F}(\mathbf{u}^{+}_{i+\frac{1}{2},j}) - \mathbf{F}(\mathbf{u}^{-}_{i+\frac{1}{2},j})\right)\right] \tag{6}$$

where $\alpha_{\mathbf{u}}$ is the associated Lax-Friedrichs global maximum characteristic speed. Now, we discretize Eq. 4 explicitly as:

$$\frac{\mathbf{u}^{n+1}_{i,j} - \mathbf{u}^{n}_{i,j}}{\Delta t} + \frac{\mathbf{F}^{n}_{i+\frac{1}{2},j} - \mathbf{F}^{n}_{i-\frac{1}{2},j}}{\Delta x} + \frac{\mathbf{G}^{n}_{i,j+\frac{1}{2}} - \mathbf{G}^{n}_{i,j-\frac{1}{2}}}{\Delta y} = \mathbf{S}(\mathbf{u}^{n}_{i,j}) \tag{7}$$

Note that in our case, we also choose to formulate the source term $\mathbf{S}(\mathbf{u}^{n}_{i,j})$ explicitly and centered at the grid point. Since we use an entirely explicit formulation, we can rewrite Eq. 7 as a time stepping operator for $\mathbf{u}^{n+1}$, namely $\mathbf{u}^{n+1} = \mathbf{T}(\mathbf{u}^{n})$. Because Runge-Kutta uses multiple stages within each time step, we reassign the output of the $\mathbf{T}$ operator to be $\mathbf{u}^{(n+1)} = \mathbf{T}(\mathbf{u}^{n})$, where $(n+1)$ indicates an intermediate full time step forward. This means a full Runge-Kutta step progresses as follows:

$$\mathbf{u}^{(n+2)} = \mathbf{T}(\mathbf{T}(\mathbf{u}^{n})) \quad \longrightarrow \quad \mathbf{u}^{(n+\frac{3}{2})} = \mathbf{T}(0.25\mathbf{u}^{(n+2)} + 0.75\mathbf{u}^{n}) \quad \longrightarrow \quad \mathbf{u}^{n+1} = \frac{2}{3}\mathbf{u}^{(n+\frac{3}{2})} + \frac{1}{3}\mathbf{u}^{n} \tag{8}$$

The process outlined by Eq. 8 outputs a final $\mathbf{u}^{n+1}$ representing a full-step forward in simulation time.

## TPU implementation

To leverage the TPU's several cores, we divide the domain into multiple subdomains and independently compute the numerical solution to the governing equations on each core. While a lot of the computation can take place independently, each subdomain remains dependent on the others via their boundaries and the Lax-Friedrichs global maximum in characteristic speed. We determine global maximum characteristic speed by sharing and reducing the Lax-Friedrichs maximum characteristic speed calculated on each core. We transfer subdomain boundary information with further care by using a halo exchange. The data transfer behavior and computation structure is summarized in Fig. 3.

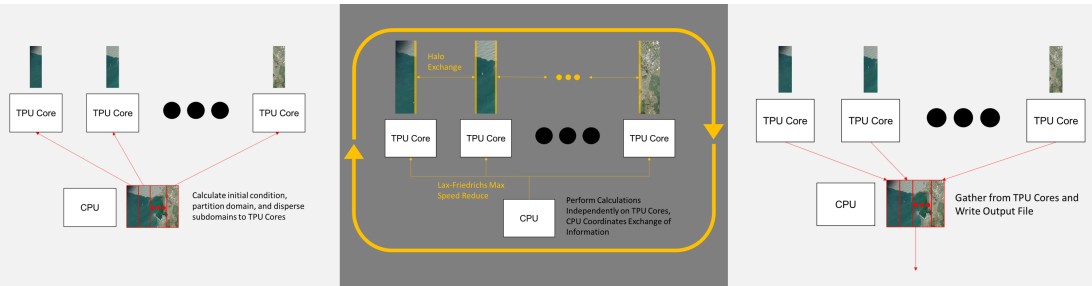

**Figure 3.** Left: Initialization of implementation takes advantage of CPU to allocate initial conditions and topography. Center: Regular computation period occurring on each subdomain, run independently on TPU cores with some data sharing coordinated by CPU. Right: CPU Gather to write results to output files.

Our implementation is inspired by Hu et al. (2022), who chose halo exchange as an instrument for the TPU to communicate information across subdomain boundaries in their formulation of the shallow water equations. In the halo exchange process, we transfer slices of the domain from one core to the others immediately adjacent. While Hu et al. (2022)'s methodology only involved the exchange of a single slice from one core to the other, we transfer several slices in order to take full advantage of the high accuracy and larger footprint of the WENO scheme. These halo exchanges are then performed in every stage of the Runge-Kutta scheme, meaning that they occur multiple times in a single time step.

The initial conditions and results are communicated from the remote program, which resides on the CPU, to the TPU workers by means of `tpu.replicate` which sends `TensorFlow` code to each TPU. We refer to Hu et al. (2022) for further details on the TPU implementation.

## 2 Model verification and validation

We differentiate between model verification and validation in the manner suggested by Carson (2002). Specifically, we check for model and implementation error by quantifying the extent to which numerical solutions compare to correct analytical solutions (Carson, 2002): wet dam break (Section 2.1), oscillations in a parabolic bowl (Section 2.2), and steady state flow down a slope with friction (Section 2.3). Following this, we validate by checking how well numerical solutions reflect the real system and apply to the context (Carson, 2002). To do this, we compare against an existing numerical benchmark from the Inundation Science and Engineering Cooperative (ISEC, 2004) and results from an investigation of nature-based solutions (Lunghino et al., 2020) in Section 2.4, and consider the propagation of a computed tsunami over the observed topography of Crescent City in Section 2.5.

To quantify the accuracy of the solutions, we test our numerical solver against some classical analytical solutions to the shallow water equations. We assess the model's ability to capture key physical processes relevant to inundation, including steep wave propagation, friction, and topography dependence. We use relative errors in the $L_\infty$ and $L_2$ sense as the metric to determine model accuracy. These are approximated in this paper in the following manner:

$$L_\infty = \frac{\max_\Omega |h_c - h_a|}{\max_\Omega |h_a|}, \quad L_2 = \sqrt{\frac{\sum_\Omega (h_c - h_a)^2}{\sum_\Omega (h_a)^2}}, \tag{9}$$

where $h_c$ is the computed solution at the discretized cells, $h_a$ is the analytical solution at the corresponding cells, and $\Omega$ denotes the computational domain. We omit the cell sizes that often serve as weights in the $L_2$ relative error norm because all of our cells have the same size. The $L_2$ relative error norm indicates the sum total error in water levels throughout the entire domain, while the $L_\infty$ norm indicates the maximal error in a single cell's water level compared to the actual solution.

We refer interested readers to the Appendix for the corresponding grid convergence analysis under these relative error norms for the first three analytical cases, and we refer readers to Section 3.2 for grid convergence analysis of the tsunami modeling in the context of the ISEC benchmark as well as the Crescent City scenario.

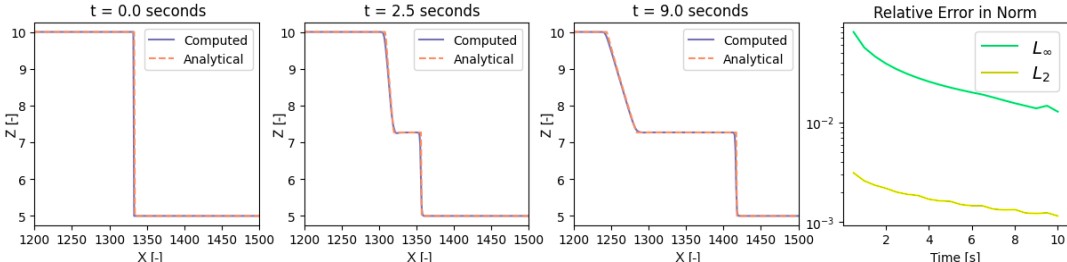

**Figure 4.** On the left, several instances in time of the computed (purple) water heights to wet dam break compared with the analytical (orange, dashed) water heights. The rightmost figure plots the $L_2$ and $L_\infty$ relative norms of the error between the analytical and computed solutions.

## 2.1 Wet dam break

The classical one-dimensional Wet Dam Break (Stoker, 1957) provides us an opportunity to test the ability of our code to capture shock propagation and advection. In this case, there is no friction ($n = 0$) and the topography is flat ($b(x) = 0$). The boundaries are set at a constant height with zero flux. We impose the following initial condition:

$$(hu) = 0, (hv) = 0, h(x) = \begin{cases} h_l & x \le x_0 \\ h_r & x > x_0 \end{cases}, \tag{10}$$

200

where $h_l$ and $h_r$ are the constant water heights on either side of a shock front $x_0$. We compare our numerical solution for water height against the dynamic analytical solution from Delestre et al. (2013):

$$205 \quad h(x,t) = \begin{cases} h_l & x \leq x_1 \\ \frac{4}{9g}\left(\sqrt{gh_l} - \frac{x-x_0}{2t}\right)^2 & x_1(t) < x \leq x_2(t) \\ \frac{c_m^2}{g} & x_2(t) < x \leq x_3(t) \\ h_r & x > x_3(t) \end{cases}, \tag{11}$$

$$x_1(t) = x_0 - t\sqrt{gh_l}, \tag{12}$$

$$x_2(t) = x_0 + t(2\sqrt{gh_l} - 3c_m), \tag{13}$$

$$x_3(t) = x_0 + t\frac{2c_m^2(\sqrt{gh_l} - c_m)}{c_m^2 - gh_r}, \quad \text{and} \tag{14}$$

$$c_m \text{ is the solution to } -8gh_r c_m^2(\sqrt{gh_l} - c_m)^2 + (c_m^2 - gh_r)^2(c_m^2 + gh_r) = 0. \tag{15}$$

A qualitative comparison of the computed and analytical solutions for times $t = 0$, 2.5, and 9 seconds is shown in the left plots of Fig. 4. The relative error between the analytical and computed solutions in the infinity and 2-norms at a small distance away from the shock front are plotted on the right. We interpret the converging relative error norms to a low magnitude as verification of our implementation to sufficiently capture shock propagation and advection.

### 2.2 Planar parabolic bowl

The classical one-dimensional planar parabolic bowl originally suggested by (Thacker, 1981), is an oscillating solution allowing us to test the source term for topography without friction ($n = 0$). We enforce homogeneous Dirichlet conditions in both flux and water height, at a resolution of 1 m. Once again, we take the test directly from Delestre et al. (2013), resulting in the following description of the base topography:

$$b(x) = h_0\left(\frac{1}{a^2}\left(x - \frac{L}{2}\right)^2 - 1\right), \tag{16}$$

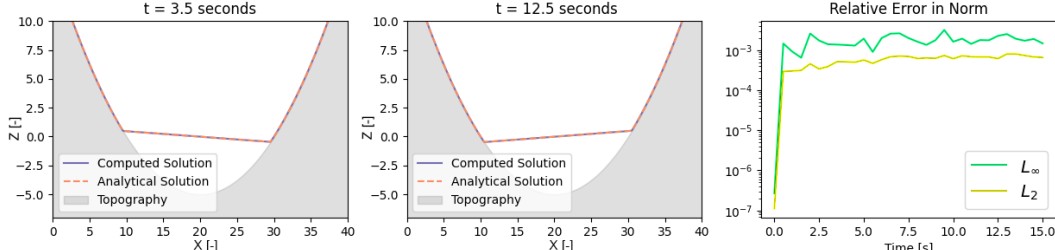

**Figure 5.** On the left, several instances in time of the computed (purple) water heights to the one-dimensional parabolic bowl compared with the analytical (orange, dashed) water heights. The rightmost figure plots the $L_2$ and $L_\infty$ relative norms of the error between the analytical and computed solutions.

corresponding with the following initial condition:

$$(hu) = 0, (hv) = 0, h(x) = \begin{cases} -h_0 \left( \left( \frac{2x-L+1}{2a} \right)^2 - 1 \right) & \frac{1-2a+L}{2} < x < \frac{1+2a+L}{2} \\ 0 & \text{otherwise} \end{cases} . \tag{17}$$

This leads to the following dynamic analytical solution for the water height:

$$h(x,t) = \begin{cases} -h_0 \left( \left( \frac{2x-L}{2a} + \frac{1}{2a} \cos \left( \frac{\sqrt{2gh_0}t}{a} \right) \right)^2 - 1 \right) & x_1(t) < x < x_1(t) + 2a \\ 0 & \text{otherwise} \end{cases} , \tag{18}$$

where $x_1(t) = \frac{1}{2} \cos \left( \frac{\sqrt{2gh_0}t}{a} \right) - a + \frac{L}{2}$. A qualitative comparison of the parabolic bowl solution at the time instances $t = 3.5$ sand $t = 12.5$ s can be seen on the left of Fig. 5. The analytical and computed solutions appear to correspond to one another well. For a more quantitative analysis, the relative error-norms of the solutions are depicted on the right of Fig. 5. We interpret the converging relative error norms to a low magnitude as verification of our implementation to sufficiently capture the source term of the shallow water equations induced by topography.

### 2.3 Steady flow down a slope with friction

We do a short test in order to assess the correctness the discretized friction source term, focusing on a relatively simple flow down a slope with finite friction ($n = 0.033$) as tested by Xia and Liang (2018).

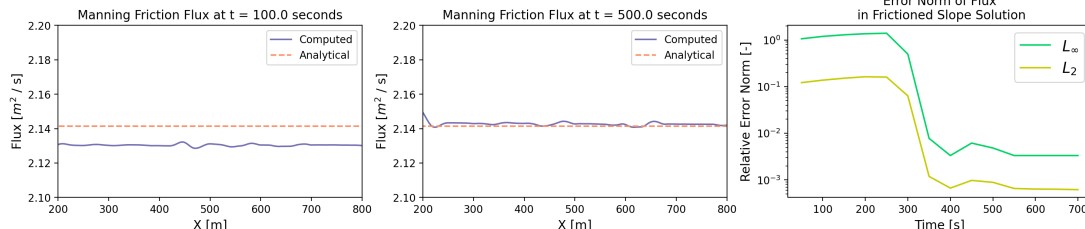

**Figure 6.** On the left, several instances in time of the computed (purple) fluxes given a water level and a slope, compared to the analytical (orange, dashed) flux. The rightmost figure plots the $L_2$ and $L_\infty$ relative norms of the error between the analytical and computed solutions.

The steady state flow down a slope then becomes

$$(hu) = \frac{\sqrt{b_x}}{n} h^{\frac{5}{3}} \tag{19}$$

where $b_x$ is the slope. In this test, we initialize the problem wave height of 0.5 and a slope of $\frac{1}{20}$, while allowing the flux to start at zero. This specific example and its convergence toward steady state is shown in Fig. 6. The left plots show the flux at $t = 100$ s and $t = 500$ s. We see that the flux rises from zero

towards the steady state flux level. On the right plot, the error norm of the steady state flux takes some time to reach steady state, but reaches a very small level upon reaching time $t = 500$ s. Because we approach the appropriate steady state solution and achieve a very small error norm, our implementation is verified in capturing a manning friction law.

## 2.4   Validation for tsunami simulations

To assess the ability of the code to capture tsunami propagation, we start with a popular numerical benchmark from the Inundation and Science Engineering Cooperative (ISEC) (ISEC, 2004) that represents tsunami runup over an idealized planar beach that provides solutions for tsunami runup at times $t =$ 180 s, 195 s, 220 s. We formulate the initial condition for water height using Lunghino et al. (2020). The solutions from the benchmark (dashed, orange) are qualitatively compared with the numerical so-

lution produced by our code (solid, purple) in Fig. 7. We take the qualitative agreement as validation of the model's ability to model the runup of a Carrier N-Wave (Carrier et al., 2003).

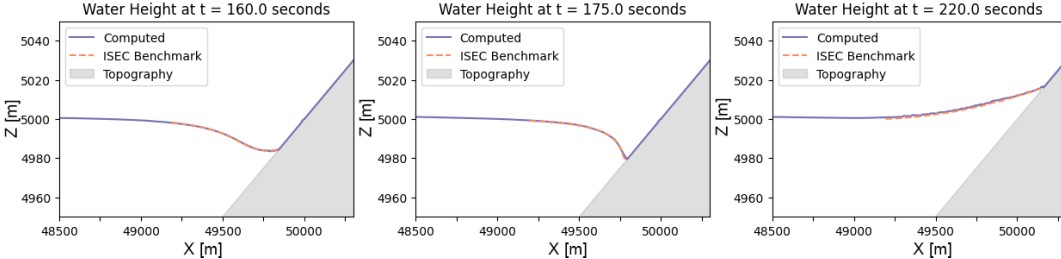

**Figure 7.** Qualitative comparison of the computed solution with resolution 1 m compared to the ISEC benchmark at three different time instances.

Since we are interested in leveraging TPUs for tsunami-risk mitigation planning, we take a look at the ability of our shallow water equation code to reproduce a few particular results by Lunghino et al. (2020) who investigated the effects of hills on a tsunami running up on a planar beach. The tsunami is initialized as Carrier's N-wave (Carrier et al., 2003):

$$\eta = 2(a_1 \exp\{-\hat{k}_1(x - \hat{x}_1)^2\} - a_2 \exp\{\hat{k}_2(x - \hat{x}_2)^2\}), \tag{20}$$

where $\eta = h + z$, $\hat{x}_1 = 1000 + 0.5151125\lambda$, $\hat{x}_2 = 1000 + 0.2048\lambda$, $\hat{k}_1 = 28.416/\lambda^2$, $\hat{k}_2 = 256/\lambda^2$, $a_1 = A$, and $a_2 = A/3$. While this is the analytically correct form, the flow origin in the code is not the shoreline, so there are some effective shifts $\hat{x}_1$ and $\hat{x}_2$ that we need to do. An example of the Carrier wave initial condition and offshore propagation behavior for $A = 15$ m and $\lambda = 2000$ m is shown in Figure 8. We apply free slip, no-penetration boundary conditions to the four domain boundaries, which means that the component of the boundary-normal component of the velocity vector is zero whereas its tangential component is unaltered. The shallow water equation model presented in this study is able to reproduce the wave reflection provided by a hill, consistent with results from Lunghino et al. (2020). Because this simulation is possible by the implementation, other further analysis can be conducted to understand the mitigative benefit of other nature-based solutions.

## 2.5 Real-world scenario

Past tsunamis impacting the West Coast of the United States have caused more damage around the harbor of Crescent City in California than elsewhere along the Pacific Coast (Arcas and Uslu, 2010).

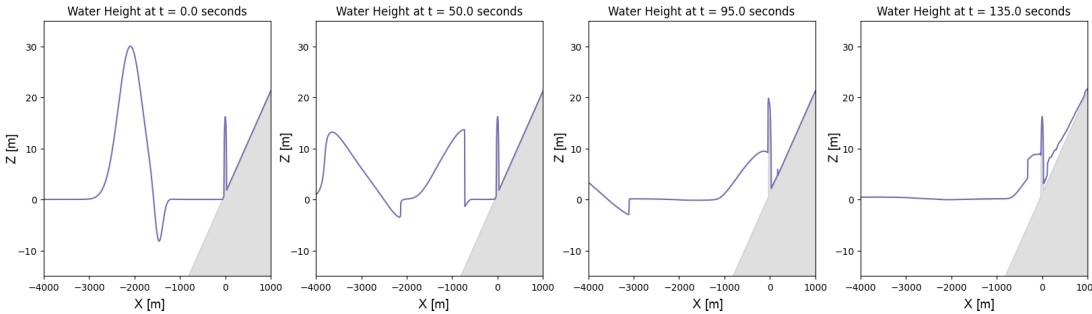

**Figure 8.** Several snapshots in time of the tsunami's propagation over a modeled ellipsoidal hill on a slope. From left to right, the formulation of the initial Carrier N-wave at $t = 0$, followed by the propagation of a wave front toward the hill at $t = 50$, collision of the wave front of the hill at $t = 95$, and the formation of a reflected wave at $t = 135$.

For this reason, we chose an area of approximately 105 km$^2$ around Crescent City to demonstrate the code's ability to capture the impact of an idealized tsunami event for a real location at high resolution. To approximate the actual bathymetry and topography, we use a Digital Elevation Model for this area with uniform grid spacing of 4 m provided by NOAA (NOAA National Geophysical Data Center, 2010; Grothe et al., 2011). For the sake of providing a proof of concept, we initialize the tsunami using the same Carrier's waveform defined above but with the following parameters: $A = 10$ m, $\lambda = 2000$ m, $\hat{x}_1 = 6000 + 0.5151125\lambda$ m, and $\hat{x}_2 = 6000 + 0.2048\lambda$ m. The chosen parameters lead to maximum inundation patterns similar to that seen in one modeled extreme scenario from Arcas and Uslu (2010). In Fig. 9, we start with an absence of any nearshore wave (including at $t = 50\ s$) and then a development of a tsunami front that is visible to the shoreline by $t = 140\ s$. That front penetrates the harbor by $t = 220\ s$, and is soon followed by the inundation of the coastline as well as reflection of wave energy back to the ocean. We also observe that the mountain range on the upper part of the figure clearly provides a significant protective benefit to the land beyond it.

The protective benefit of the mountain range can be further seen in Fig. 10. This high-water map from a 10-min simulation of runup due to the Carrier N-wave shows the spatial variation of which locations see at least 1 m of inundation under different wave amplitudes. While more work would be necessary to leverage our tsunami software package to connect the inundation by a generated wave

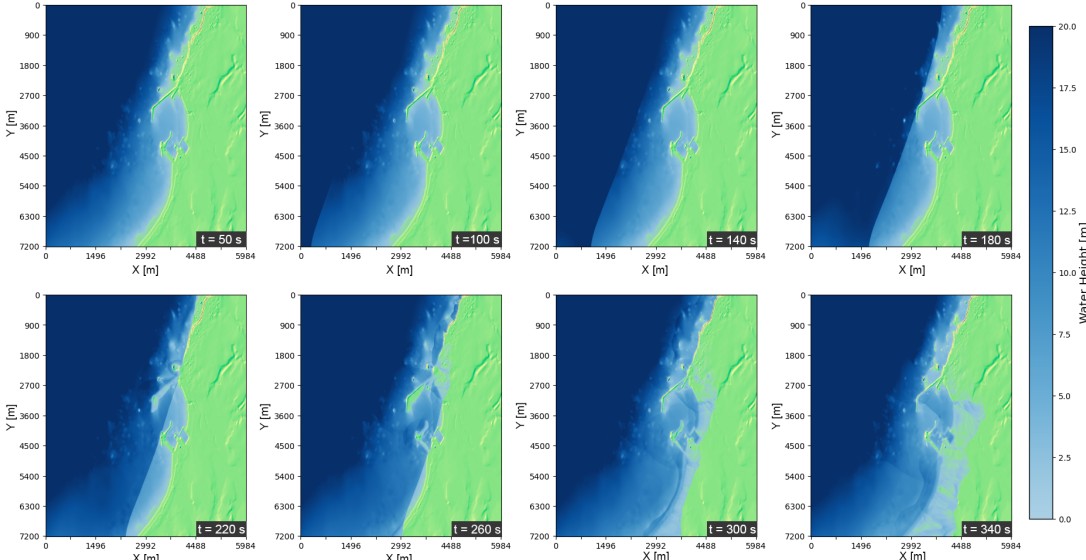

**Figure 9.** Several snapshots of modeled tsunami propagation over terrain and geological features of Crescent City, CA, where any level of blue indicates water cover and green depicts a stylized map of the topography above surface level. From left to right, then top to bottom, we have steady state near shore at $t = 50\ s$; followed by the propagation of a wave front at $t = 100\ s$ and $140\ s$; contact with Crescent City harbor at $t = 180\ s$; inundation of the harbor and some of the coastline at $t = 220\ s$ and $260\ s$; and tsunami reflection and inundation at $t = 300\ s$ and $340\ s$.

to the forcing generated by earthquakes, a risk manager can see how higher magnitude tsunamis may disproportionately affect certain locations over others.

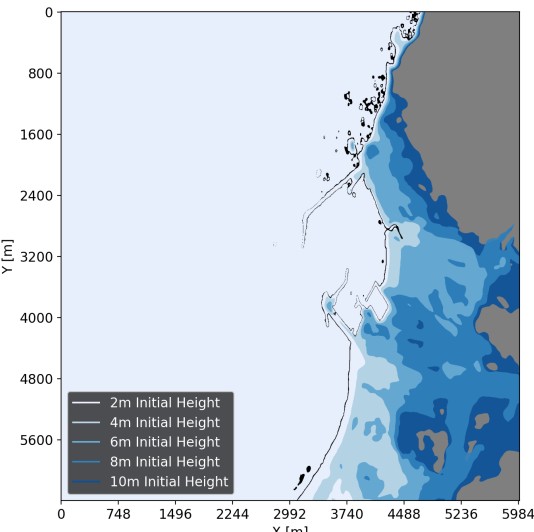

**Figure 10.** 1 m inundation High-water map for Crescent City, CA, under different amplitude Carrier N-waves initialized 1 km offshore. 10 minutes of simulated time. Darker shades of blue indicate the extent reached by higher amplitude N-waves.

## 3 Performance analysis on TPU

### 3.1 Number of TPU cores

As noted in the introduction, in communities where users may not have access to high performance computing facilities, the Cloud TPU Platform provides a particularly valuable resource where users can perform large-scale computations rapidly. To quantify the potential speed-up enabled by TPUs with increasing numbers of cores, we observe the average wall-clock time taken in computation for each time-step with the exclusion of the first time-step. This first time-step includes several preprocessing functions, such as reading DEM files into TPU memory, setting up initial conditions, and initializing the Tensorflow workflow. Similarly, we calculate runtime based on the amount of time spent in computation with the exception of this first step, with time which is variable from run-to-run. As shown in Table 1, the problem size posed by the realistic scenario is sufficient to see rapid improvements in runtime based

on the number of cores. We note that our analysis may vary user-to-user, depending on the TPU version that they are allocated and the number of cores available to them. Our simulations were all conducted with a TPUv2, and we extend our analysis only up to 8 TPU cores because, at the time of writing, Google Colab only provides 8 cores for free in our region.

| Number of Cores | 1 | 2 | 4 | 8 |
|---|---|---|---|---|
| Average Runtime / Time-step [ms] | 25.8 | 16.5 | 9.6 | 6.4 |
| Speedup over 1 Core [-] | * | 1.6 | 2.7 | 4.03 |

**Table 1.** Average TPU Runtime per time-step (in milliseconds) with varying numbers of TPUv2 cores. The Crescent City configuration at an 8 m resolution is used, with time-steps of $\Delta t = 0.02$ s for a total of 400 simulated seconds. The domain is a grid of approximately 901 by 1992 elements; TPU cores find solutions to subdomains divided in the y-direction as suggested by Hu et al. (2022) and graphically depicted in Fig. 3

### 3.2 Geophysical problem resolution

Simulating tsunami runup typically requires large domains and sufficiently high resolution to accurately capture tsunami propagation and inundation over complex topography. Therefore, we continue analyzing our Crescent City scenario for both convergence and the average runtime spent for each time-step under varying degrees of resolution, shown in Fig. 11 and with numerical results given in Table 2. As is expected, relative error norms fall as we reach higher resolutions and lower cell sizes, which in turn corresponds to increasing runtime spent for each time-step. To expand upon the scaling of runtime, we calculate an accessory 'efficiency' metric to more specifically understand how runtime scales with the number of elements, calculated by dividing the total number of elements over the average runtime per time-step. Higher values for this efficiency metric would be associated with a greater number of computations for grid elements being completed per unit time. We see that this efficiency is maximized for high numbers of elements as expected while the TPU reaches capacity.

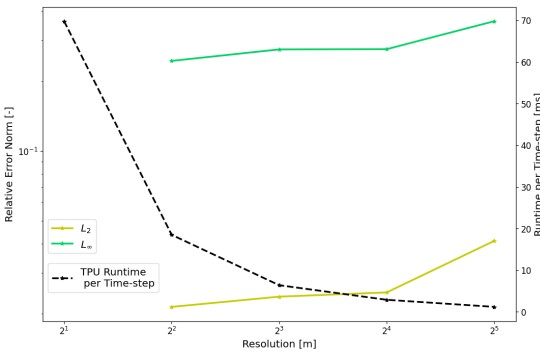

**Figure 11.** Average TPU runtimes per time-step (black) and relative $L_2$ (gold) and $L_\infty$ (green) norms under varying resolutions for computing the tsunami propagation over the Crescent City DEM.

| Resolution [m] | 2 | 4 | 8 | 16 | 32 |
|---|---|---|---|---|---|
| Average Runtime/Time-step [ms] | 69.8 | 18.5 | 6.4 | 2.9 | 1.2 |
| Number of Elements [millions] | 29 | 7.2 | 1.8 | 0.45 | 0.11 |
| Efficiency [elements / ms] | 4.1E5 | 3.8E5 | 2.8E5 | 1.6E5 | 9.1E4 |
| Relative $L_2$ Error Norm | * | 0.0214 | 0.0236 | 0.0247 | 0.0411 |
| Relative $L_\infty$ Error Norm | * | 0.245 | 0.274 | 0.275 | 0.363 |

**Table 2.** Approximate TPU Runtimes (in seconds) for a 400 second simulation with varying resolutions of the Crescent City Configuration using time step of $\Delta t = 0.02$ s. We compute $L$ Error norms against the 2 m resolution for correctness at times t = 100, 200, 300, and 400 seconds in the coastal region. All simulations ran on a single TPUv2 with 8 cores.

We perform the same analysis under varying degrees of resolution using the benchmark from the Inundation Science and Engineering Cooperative (ISEC, 2004) that we previously validated against in Section 2.4. We show a qualitative comparison of the tsunami propagation under different resolutions are graphically depicted in Fig. 12 in the top two and bottom left figures. In the bottom right plot of Fig. 12, we see the expected fall in runtime based on coarser resolution (black), and a rise in relative error, an error representing the accumulation of all three time instances for which the ISEC benchmark is

defined. We provide an overview of the corresponding values and efficiency metrics in Table 3. Since the total number of elements is smaller for the ISEC benchmark as compared to the Crescent City scenario, we are able to see even lower efficiency as the TPU is even further away from reaching full capacity.

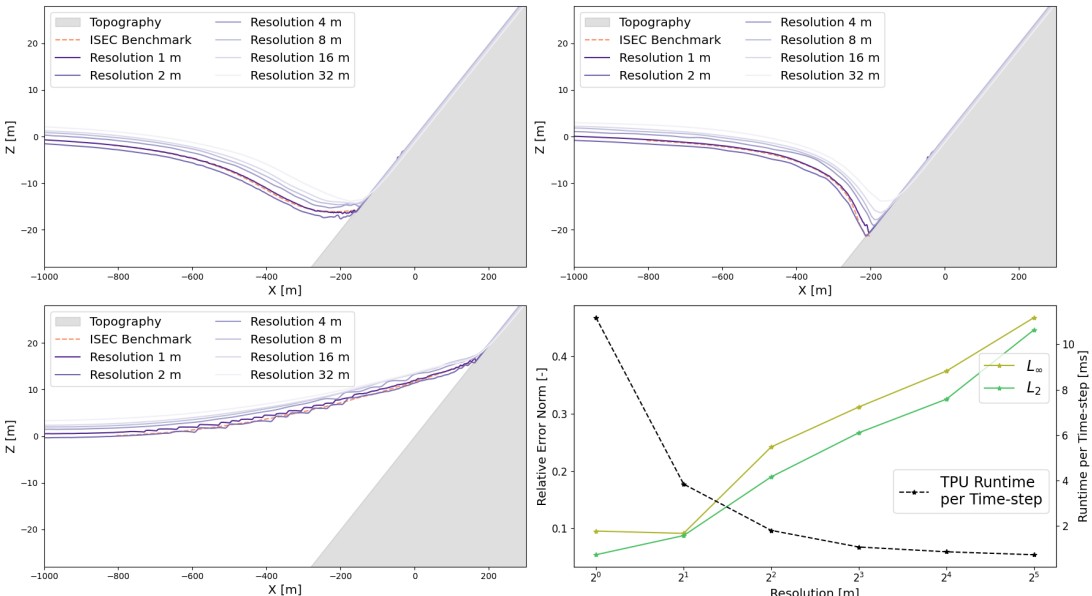

**Figure 12.** Top left: ISEC Benchmark comparison at 160 seconds. Top right: ISEC Benchmark comparison at 175 seconds. Bottom Left: ISEC Benchmark comparison at 220 seconds. Bottom Right: TPU runtime for each time-step (black) and corresponding relative $L_2$ (green) and $L_\infty$ (gold) error norms for varying simulation resolutions.

| Resolution [m] | 1 | 2 | 4 | 8 | 16 | 32 |
|---|---|---|---|---|---|---|
| Average Runtime/Time-step [ms] | 11.17 | 3.85 | 1.81 | 1.09 | 0.87 | 0.74 |
| Number of Elements [millions] | 5.1 | 1.3 | 0.33 | 0.082 | 0.022 | 0.006 |
| Efficiency [elements / ms] | 4.5E5 | 3.3E5 | 1.8E5 | 7.5E4 | 2.5E4 | 8.5E3 |
| Relative $L_2$ Error | 0.054 | 0.087 | 0.190 | 0.267 | 0.325 | 0.446 |
| Relative $L_\infty$ Error | 0.095 | 0.091 | 0.242 | 0.312 | 0.374 | 0.468 |

**Table 3.** Average TPU runtimes per time-step (in ms) under varying resolutions for the ISEC Tsunami Benchmark using time step of $\Delta t = 5 \cdot 10^{-3}$ s, along with corresponding $L$-norms calculated with respect to all three times where the ISEC benchmark was available. All simulations ran on a single TPUv2 with 8 cores.

**3.3 Comparison with GeoClaw**

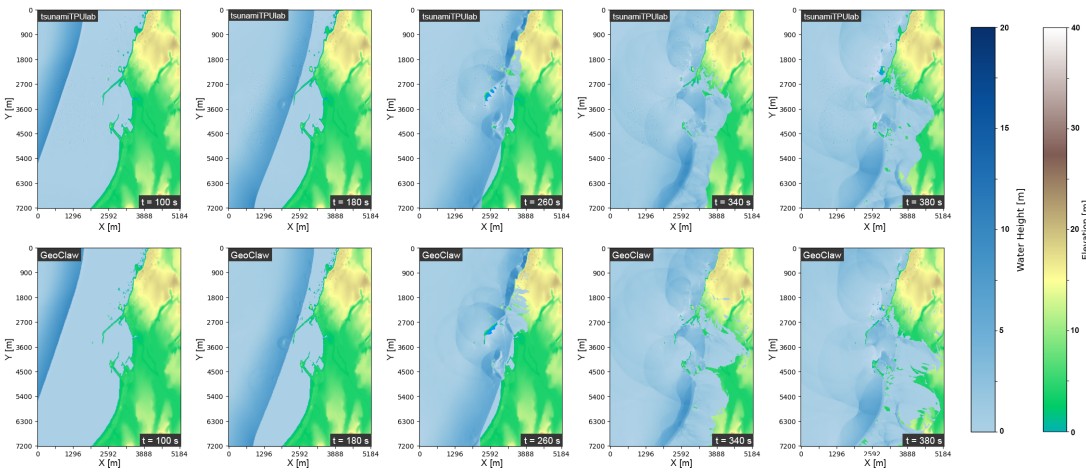

**Figure 13.** TPU solution (top row) at several time instances compared to the GeoClaw solution (bottom row). The arrival of the tsunami front ($t = 100, 180$ s), the inundation of the harbor ($t = 260$ s), and coastal inundation and reflection is depicted, and relatively comparable.

For comparison purposes, we run GeoClaw (Clawpack Development Team, 2020; Mandli et al., 2016; Berger et al., 2011) using 4 CPU threads that we were allocated for free via Google Colab. Our GeoClaw simulation uses the same DEM file and is computed at an 8 m resolution without mesh-refinement. We use an adaptive time-step bounded by the Courant–Friedrichs–Lewy condition, and determine the

330 spatial fluxes using a second-order, rate-limited Lax-Wendroff scheme. In Fig. 14, we see lower relative error norms as we approach higher resolution in our simulation of our Crescent City scenario. The GeoClaw numerical solution can be compared to our TPU numerical solution in Fig. 13, where the top row includes several instances in time of the TPU numerical solution, and the bottom row depicts the GeoClaw numerical solution at the same instances in time. Although some differences can be seen

in inundation by $t = 380\ s$ in the rightmost plots, the solutions do generally appear similar over time, lending credibility to the validity of the numerical solution presented in this paper.

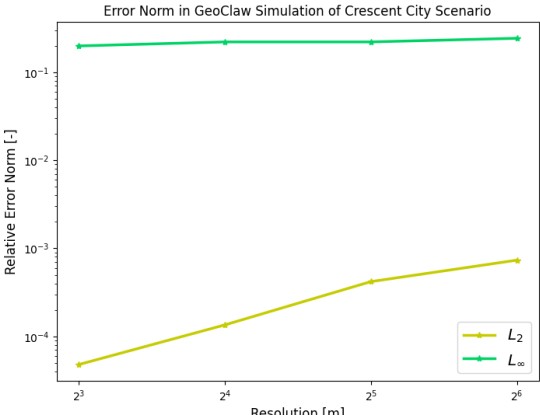

**Figure 14.** GeoClaw relative error norms for a 100 s simulation for our Crescent City Scenario under varying resolutions. 4 m resolution is used as the benchmark.

One particular advantage of our TPU-based code lies in its comparatively rapid simulation. While our TPU-based code completes a 400 second simulation in approximately 2 minutes of wall-clock time, the adaptive-time-stepping GeoClaw implementation on the CPU takes approximately 275 minutes. The

340 TPU sees even more significant speedup when we enforce a fixed time-step equal to that of the TPU implementation: that GeoClaw simulation would take over 1300 minutes.

### 3.4 Energy utilization

Estimates of energy efficiency of computing operations are becoming increasingly popular, especially in response to progressing climate change (Fuhrer et al., 2018; Fourestey et al., 2014). To get a rough approximation of the comparative efficiency of a Google Cloud TPU over a CPU, we ran the Crescent City tsunami propagation problem at 8 m resolution using GeoClaw on a Google Cloud CPU node and our code on a Google Cloud TPU node. We do not have the information nor access to the physical devices needed to conduct rigorous energy profiles to calculate efficiency as is done by others (e.g., Ge et al., 2010), so we deliver an order-of-magnitude estimate based on the Thermal Design Power of the devices that we are allocated via Google Colab. We only compared resources that are freely available through Colab in order to compare the efficiencies of computing resources that may be accessible to all users.

Based on Table 2, each time step takes an average of about $6.4 \times 10^{-3}$ seconds on the TPUv2 that we were allocated by Google Colab, corresponding to about 0.32 seconds per simulated second under our current time-stepping regime. The TPUv2 we were assigned contains 4 chips with a Thermal Design Power of 280 W per chip (Jouppi et al., 2021), meaning that each simulated-second then has an energy cost of approximately 0.1 Wh, leading to an approximately 40 Wh energy cost for a 400 simulated-second simulation. At a price of 21 cents/kWh in the U.S. at the time of writing this article, this simulation has a monetary cost of 0.84 cents.

When we ran GeoClaw for our CPU comparison on energy utilization, we enforced a fixed time step on the GeoClaw package of equal size to that of our TPU, i.e., $\Delta t = 0.02$ s, rather than leveraging GeoClaw's adaptive time-stepping to have a fair comparison in terms of the approximate number of operations. While the specific processor we were allocated for this CPU comparison is not immediately clear, we deduced from the model name, CPU family, and model number that we were allocated the Intel Xeon E5-2650 v4 with a base frequency of 2.2 GHz and a Thermal Design Power of 105 W for 12 cores and 24 threads (Intel, 2016). Of this, we were allocated 2 cores/4 threads, and we took full advantage of all threads for our GeoClaw run. Each time-step took approximately 4.2 seconds, corresponding to about 205 runtime seconds per simulated second. If we assume ideal conditions leading to perfectly proportional scaling of computational speed in increasing cores, we would expect that each simulated second would take approximately 34.2 seconds when the Intel Xeon E5-2650 v4 was used

to full capacity. This means that each simulated second would have an associated energy cost on the order of 1.0 Wh. A 400 modeled-second simulation would imply a total cost of approximately 400 Wh of energy, or a monetary cost of 8.4 cents. With the corresponding TPU energy calculation in mind, our conservative estimate suggests that a CPU simulation has approximately 10 times the energy cost of running an equivalent TPU simulation under the same time-stepping conditions.

While these two simulations accomplish the same thing, they have vastly different associated performances. At times, rapid computation and simulations are necessary in the context of risk analysis, and the associated energy costs of such a performant computation is worth estimating. To address this, we push our energy estimate a touch further, providing another order-of-magnitude estimate of what a CPU simulation conducted at TPU performance would be. We extrapolate our previous assumptions further, assuming proportional scaling of computational speed with increasing CPUs, and that the Thermal Design Power applies to each CPU within a system independently. Because a simulated second of a full capacity Intel Xeon E5-2650 v4 CPU takes approximately 34.2 seconds compared to the TPUv2's 0.32 seconds, over 100 CPUs at full capacity would be needed for similar rapidity in simulation. Following similar logic as done in the previous paragraph, a CPU simulation of equivalent performance would have approximately 1000 times the energy cost of running a TPU simulation when ran under the same time-stepping conditions.

## 4 Discussion

Sustainable tsunami-risk mitigation in the Pacific Northwest is a challenging task. Some challenges come from beneath, because previous large subduction zone earthquakes at Cascadia led to $0.5 - 1$ m of co-seismic subsidence, the sudden sinking of land during an earthquake (Wang et al., 2013). Strong shaking can also lead to liquefaction (Atwater, 1992; Takada and Atwater, 2004). Other challenges come from the ocean, where sea-level rise (Church and White, 2006; Bindoff et al., 2007) and intensifying winter storms (Graham and Diaz, 2001) have increased wave heights (Ruggiero et al., 2010; Ruggiero, 2013) and accelerated coastal erosion (Ruggiero, 2008). A recent USGS report documented rapid shoreline changes at an average rate of almost 1 m/yr across 9,087 individual transects (Ruggiero et al., 2013), suggesting the possibility that the shoreline might change significantly during the century-long return-period of large earthquakes in Cascadia (Witter et al., 2003).

The picture that emerges is that of a highly dynamic coastline – maybe too dynamic for an entirely static approach. Nature is not only continuing to shape the coastline, but is also a fundamental component of the region's cultural heritage, identity and local economy. So, it is maybe not surprising that the Pacific Northwest is a thought-leader when it comes to designing hybrid approaches to sustainable climate adaptation through the Green Shores program (Dalton et al., 2013) and to vertical tsunami evacuation through Project Safe Haven (Freitag et al., 2011).

Project Safe Haven is a grass-roots approach to reducing tsunami risk mostly by providing accessible vertical-evacuation options for communities. Many proposed designs entail reinforced hillscapes like the one shown in figure 2, intended to dissipate wave energy and provide vertical evacuation space during tsunami inundation. To build confidence in such a solution and its mitigation effects, risk managers must be able to quickly and precisely forecast a tsunami inundation, preferably via a publicly available, centralized modeling infrastructure.

This paper aims to be a first step towards a community based infrastructure that will allow local authorities around the world to readily execute tsunami simulations for risk mitigation planning. We aim to provide a proof-of-concept rather than a complete implementation. As such, we used a very similar base framework used by Hu et al. (2022) of halo exchange in combination with a WENO (Liu et al., 1994; Jiang and Shu, 1996) and Runge-Kutta scheme (Shu, 1988). We choose easily implementable higher order schemes to maintain high accuracy, necessary for simulating tsunami inundation over the complex topographies that risk managers often deal with in the real-world, despite the fact that the large stencils within the current implementation may not be optimal for TPU performance. Future work could consider a convolution-based implementation of the quadrature of the shallow water equations to test for maximum performance utilization of the TPUs.

Because our code is specifically an implementation of the shallow water equations, it is currently unable to model tsunami initiation, or any fluid structure interactions that may be desired to accompany analysis of nature-based solutions. Instead, it requires an initial condition for wave heights and fluxes, meaning a full tsunami simulation would require coupling the results of a tsunami initiation model as an input. While our implementation is relatively limited in scope, the model is able to provide a starting point for a more complete software package for communities as they evaluate nature-based options for tsunami mitigation.

We argue that Cloud TPUs are preferable to large, heavily parallel simulations on CPUs or GPUs for risk managers across all communities because the TPU-based simulations we show here do not require access to the large computing clusters hosted by laboratories and universites. These clusters require huge amounts of power to run, and are usually only made available to scientists and engineers by means of competitive grants for computing time or by use of the cloud offered by private companies. However, an expert user knowledge of these systems from a scientific computing perspective is necessary to design, run, and interpret model results, and the compute infrastructure itself may not be available to early warning centers in many parts of the world. In contrast, our code is available on Github and fully implemented in Python, can be executed through a web browser, and visualized through a simple notebook file using Google Colab without the knowledge otherwise required to run large parallel codes on high-performance computing systems. By taking advantage of Google's Cloud Platform, we also ensure that a user's power demand is met entirely with renewable energy (Google, 2022). performance can be enhanced with some knowledge about TPU architectures, community risk managers do not need this knowledge to run high quality tsunami simulations rapidly for real, physical domains with associated DEMs.

Finally, though not our focus here, we note our approach may also contribute to early tsunami warning. Once triggered, tsunamis move fast; this fact makes it necessary to model and assess their potential for damage ahead of time once they have been detected offshore. For a sufficiently fast early warning and prompt evacuation, the tsunami modeling infrastructure has an important time constraint (Giles et al., 2021) to be considered, and Faster Than Real Time (FTRT) simulations are necessary (Behrens et al., 2021; Løvholt et al., 2019). To make FTRT simulations a reality, tsunami models are being rewritten or adapted to run on Graphical Processing Units (GPUs) (Løvholt et al., 2019; Behrens and Dias, 2015; Satria et al., 2012). A TPU-based implementation as proposed here might be another meaningful step into that direction.

## 5  Conclusion

We present a first step towards an accessible software package that leverages the powers of Cloud-based TPU computing for improving the capabilities of risk managers and communities to mitigate the destructive onshore impacts of tsunamis. We verify and validate our current implementation to ensure

that it is capable of simulating inundation from a Carrier N-wave over real topography. These simulations are comparable to that ran by the popular open-source solver GeoClaw (Clawpack Development Team, 2020; Berger et al., 2011), but can be run at higher speeds through Google Colab and requires less expertise in scientific computing. As a result, high quality tsunami simulations are available to remote communities for rapidly evaluating different risk-mitigation options including but not limited to nature-based solutions. Future efforts can then be dedicated to better meeting the needs of risk managers with a platform available through the cloud, be that in coupling our shallow water equations package to earthquake-tsunami generation models, or experimenting with different numerical implementations to enable even more rapid simulation of these equations.

## Author contributions

**Ian Madden**: Methodology, Software, Analysis, Writing. **Simone Marras, PI**: Conceptualization, Methodology, Writing, Supervision. **Jenny Suckale**: Conceptualization, Methodology, Writing, Supervision.

## Competing interests

S.M. is a member of the editorial board of Geoscientific Model Development. The peer-review process was guided by an independent editor, and the authors have also no other competing interests to declare.

## Code and data availability statement

Our work is available as a GitHub release at https://github.com/smarras79/tsunamiTPUlab/releases/tag/v1.0.0 or on archive at 10.5281/zenodo.7574655.

## Acknowledgements

This work was supported by the National Science Foundation's Graduate Research Fellowships Program (GRFP) awarded to Ian Madden.

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

## Appendix

### Running the code

Due to the restrictions of the Cloud TPU using Google Cloud Storage, a user must leverage Google's buckets to run the notebooks. At the time of writing this article, 8 TPU cores are readily available in North America on Google Colab for free, but Google Cloud Storage buckets are a paid subscription service (see https://cloud.google.com/tpu/docs/regions-zones for information about specific regions). With a computing project setup on Google Cloud and a corresponding bucket with open permissions 750 (with steps specified in https://cloud.google.com/storage/docs/creating-buckets), users can quickly run any of the example notebooks or design their own simulation. Any of the example notebooks available on GitHub (with the exclusion of `tpu_tsunami.ipynb`, which contains the full implementation with all of the different scenarios; and `Create_Scenarios.ipynb`, which can aid users in generating a custom DEM file) can be quickly ran by going through the notebook after a few setup steps.

1. Download the TPU-Tsunami Repository from https://github.com/smarras79/tsunamiTPUlab/releases/ tag/v1.0.0 to your local machine. Create a project on Google Cloud Platform and associate a publicly available bucket with the project.

   2. Modify the `user_constants.py` file to specify the `PROJECT_ID` and `BUCKET` with the specifics of your Google Cloud Project. If you wish to change some simulation constants, modify

the beginning of the `tpu_simulation_utilities.py` file.

   3. Navigate to https://colab.research.google.com/ and open the example notebook (or your own notebook) from the TPU-Tsunami Repository using Colab's open from Github tool.

   4. Navigate to Runtime > Change runtime type, and verify that the TPU option is chosen as the Hardware Accelerator.

5. Upload your `user_constants.py` and `tpu_simulation_utilities.py` files to your notebook session using the drag-and-drop feature under Files. Upload any corresponding DEM files to the session as well.

   6. Specify a function corresponding to an initial condition for your DEM file (or use one example initial condition).

7. Set initial conditions, boundary conditions as clarified in the bottom of any example notebook run. Set last simulation parameters defining numerical resolution (`resolution`), time step size (`dt`), output file times, TPU core configuration (currently only capable of variation of `cy`), and DEM file name on bucket (`dem_bucket_filename`).

   8. Run the simulation.

9. Analyze results.

**Grid Convergence Analysis for Wet Dam Break (Section 2.1) and Planar Parabolic Bowl (Section 2.2)**

In addition to the convergence analysis for the ISEC Benchmark posed in Section 3.2, we have included some additional results constituting a convergence analysis for two of the other analytical scenarios that

are particularly relevant for establishing an appropriate grid resolution.

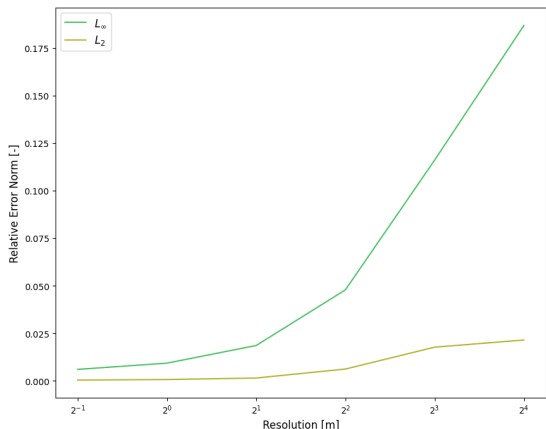

**Figure 15.** Final relative error norms calculated at 10 seconds into the wet dam break problem under varying resolutions.

| Grid Resolution | Element Count | Final $L_2$ Norm | Final $L_\infty$ Norm |
| --- | --- | --- | --- |
| 0.5 | 1.8E6 | 5.7E-4 | 7E-3 |
| 1.0 | 4.4E5 | 1.5E-3 | 0.013 |
| 2.0 | 1.1E5 | 1.8E-3 | 0.016 |
| 4.0 | 2.8E4 | 6.2E-3 | 0.047 |
| 8.0 | 7.0E3 | 0.018 | 0.12 |
| 16.0 | 1.8E3 | 0.021 | 0.187 |

**Table 4.** Grid Convergence Analysis for Wet Dam Break, with values

In Fig. 15, we can see how the we dam break case converges with higher resolution. This is more concretely specified in Table 4 with specific values. In Fig. 16, we can see improvements in the grid convergence for the planar parabolic bowl problem from section 2.2. We begin to see error convergence at high enough resolution.

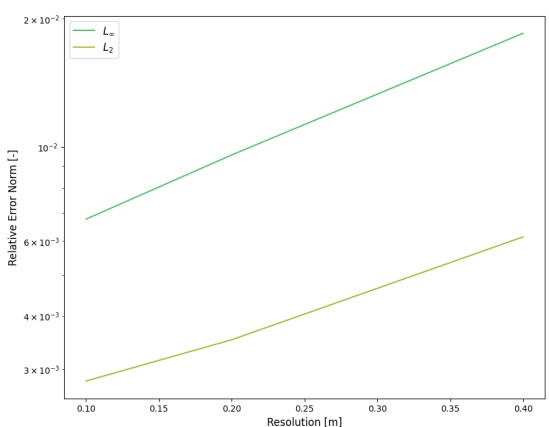

**Figure 16.** Grid Convergence Analysis for Planar Parabolic Bowl.

| Grid Resolution | Element Count | Final $L_2$ Norm | Final $L_\infty$ Norm |
|---|---|---|---|
| 0.1 | 4.4E4 | 2.8E-3 | 6.7E-3 |
| 0.2 | 1.1E4 | 3.5E-3 | 9.6E-3 |
| 0.4 | 2.3E3 | 6.1E-3 | 0.018 |

**Table 5.** Grid Convergence Analysis for Planar Parabolic Bowl.

Finally, in Fig. 17, we can see how the we dam break case converges with higher resolution. This is more concretely specified in Table 6 with specific values. In Fig. 17, we can see improvements in the grid convergence for the slope with manning friction problem from section 2.3.

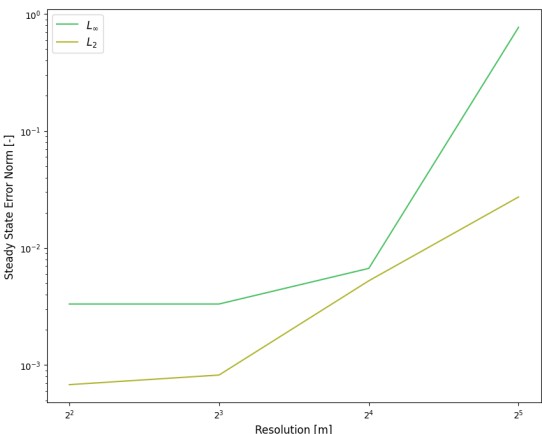

**Figure 17.** Grid Convergence Analysis using Flux as the Error Metric for Slope with Manning Friction.

| Grid Resolution | Element Count | Final $L_2$ Norm | Final $L_\infty$ Norm |
|---|---|---|---|
| 4 | 2.6E4 | 6.8E-4 | 3.3E-3 |
| 8 | 6.5E3 | 8.2E-4 | 3.3E-3 |
| 16 | 1.6E3 | 5.2E-3 | 6.7E-3 |
| 32 | 4.0E2 | 0.027 | 0.769 |

**Table 6.** Grid Convergence Analysis using Flux as the Error Metric for Slope with Manning Friction.