# Peer review of "Leveraging Google's Tensor Processing Units for tsunami-risk mitigation planning in the Pacific Northwest and beyond"

_EGUsphere, 2023_

## Author Comment (AC1)

**Manuscript Resubmission**

|  |  |
|---:|:---|
| **Journal:** | Geoscientific Model Development |
| **Manuscript #:** | EGUSPHERE-2023-116 |
| **Manuscript Title:** | Leveraging Google's Tensor Processing Units for tsunami-risk mitigation planning in the Pacific Northwest and beyond |
| **Authors:** | Ian Madden, Simone Marras, Jenny Suckale |
| **Date:** | May 5, 2023 |

Dear Dr. Deepak Subramani,

Thank you for your time overseeing the review process and for selecting reviewers that have provided insightful suggestions to improve our manuscript. It is our pleasure to resubmit our manuscript EGUSPHERE-2023-116, entitled "Leveraging Google's Tensor Processing Units for tsunami-risk mitigation planning in the Pacific Northwest and beyond" by Madden et al. for continued evaluation at Geoscientific Model Development.

In response to the comments of the reviewers, we have made the following major changes to the manuscript:

1. We have included an additional figure in section 2.5 demonstrating how a risk manager could use our TPU code as it currently stands.

2. We have added reworked the table with respect to scaling based on number of TPUv2 cores in Section 3.1.

3. Our figures and tables in Section 3.2 have been updated following a mistake in calculating error norms. We have also included an additional section to the Appendix detailing some further analysis of error norms in the problems for Section 2.1-2.3.

4. The solver conditions used in the GeoClaw software package are concretely outlined, with a grid convergence plot for the real-world scenario, in Section 3.3.

5. We have completely reworked Section 3.4 on Energy Utilization.

6. We have removed some excess summary sentences from the Discussion and added a Conclusion section.

According to the policy for resubmitting manuscripts, we have responded to all reviewer comments in detail in the below response letters.

Sincerely,
Ian Madden
Institute for Computational and Mathematical Engineering,
Stanford University, Stanford, California, USA

**Response to Reviewer #1**

*R1.1. In the equations 1–3, the non-linear advection term from page 4 seems to be contained in the term 0.5 (h2 - b2)? It would help the reader to point out this term in these equations.*

Thank you for pointing out this shortcoming to our explanation. We have added a brief sentence to remind the reader about the addition of the nonlinear advection term in our implementation of the shallow water equations on lines 118-120.

$$\frac{\partial}{\partial t}h + \frac{\partial}{\partial x}(hu) + \frac{\partial}{\partial y}(hv) = 0 \tag{1}$$

$$\frac{\partial}{\partial t}(hu) + \frac{\partial}{\partial x}\left(\frac{(hu)^2}{h} + \frac{1}{2}g(h^2 - b^2)\right) + \frac{\partial}{\partial y}(huv) = -g(h+b)\frac{\partial b}{\partial x} - \frac{gn^2\sqrt{(hu)^2+(hv)^2}}{h^{7/3}}(hu) \tag{2}$$

$$\frac{\partial}{\partial t}(hv) + \frac{\partial}{\partial x}(huv) + \frac{\partial}{\partial y}\left(\frac{(hv)^2}{h} + \frac{1}{2}g(h^2 - b^2)\right) = -g(h+b)\frac{\partial b}{\partial y} - \frac{gn^2\sqrt{(hu)^2+(hv)^2}}{h^{7/3}}(hv), \tag{3}$$

where $g = 9.81$ ms$^{-2}$ is the acceleration of gravity, and $n$ is the Manning friction coefficient. Note that the left-hand-side of our formulation of the shallow water equations includes the full nonlinear advection terms.

*R1.2. Can the authors comment further on the trade-offs of using a high-order scheme with an arguably large stencil with regard to parallel performance, numerical accuracy, and memory? This could be added to the discussion on page 22.*

Thank you for the suggestion to discuss the trade-offs of using a higher-order scheme. We added a brief explanation for the choice of numerical scheme at the end of the discussion as suggested. We do acknowledge that using a large stencil could result in reduced parallel performance; this aspect could be improved upon in the future. Here, we aim to establish an easy-to-use software package that is capable of providing a reasonable result and maintain high accuracy for the complex topographies that risk managers often deal with.

This paper aims to be a first step towards a community based infrastructure that will allow local authorities around the world to readily execute tsunami simulations for risk mitigation planning. We aim to provide a proof-of-concept rather than a complete implementation. As such, we used a very similar base framework used by Hu et al. (2022) of halo exchange in combination with a WENO (Liu et al. 1994, Jiang & Shu 1996) and Runge-Kutta scheme (Shu 1988). We choose easily implementable higher order schemes to maintain high accuracy, necessary for simulating tsunami inundation over the complex topographies that risk managers often deal with in the real-world, despite the fact that the large stencils within the current implementation may not be optimal for TPU performance. Future work could consider a convolution-based implementation of the quadrature of the shallow water equations to test for maximum performance utilization of the TPUs.

*R1.3: Can the authors give a bit more detail on the numerical treatment at shocks and at wet/dry fronts?*

For this study we assume a thin-film of water at all 'dry' points of the map. Some articles that use this approach

- P.D. Bates, J.-M. Hervouet, A new method for moving-boundary hydrodynamic problems in shallow water Proc. R. Soc. London, Ser. A: Math. Phys. Engrg. Sci., 455 (1999), pp. 3107-3128

- S. Bunya, E.J. Kubatko, J.J. Westerink, C. Dawson A wetting and drying treatment for the Runge–Kutta discontinuous Galerkin solution to the shallow water equations Comput. Methods Appl. Mech. Engrg., 198 (2009), pp. 1548-1562

- O. Gourgue, R. Comblen, J. Lambrechts, T. Kärnä, V. Legat, E. Deleersnijder, A flux-limiting wetting–drying method for finite-element shallow-water models, with application to the Scheldt Estuary, Adv. Water Resour., 32 (2009), pp. 1726-1739

- I. Nikolos, A. Delis, An unstructured node-centered finite volume scheme for shallow water flows with wet/dry fronts over complex topography, Comput. Methods Appl. Mech. Engrg., 198 (2009), pp. 3723-3750

listed by Karna Et al. (2011) "A fully implicit wetting–drying method for DG-FEM shallow water models, with an application to the Scheldt Estuary" Comput. Method Appl. Mech. Engrg.

At these dry points of the map, we use the same code provided by Hu et al. (2022) that were previously used for solving the Saint-Venant Equations on the TPU, restricting flux propagation from 'thin-film' regions. We have clarified this point in the paper.

We solve for $h$, $hu$, and $hv$ in our implementation. We further place a lower bound $h \geq \epsilon$ in all cells, meaning no properly 'dry' cells are present, and handle cells with water depth $h = \epsilon$ (the value of $\epsilon$ is problem dependent and on the order of centimeters) using the TPU code provided by Hu et al. (2022). This ensures that no flux arrives from those cells with water height $\epsilon$. While this approach has some important drawbacks (Kärnä et al. 2011), it has been used extensively (Bates & Hervouet 1999, Bunya et al. 2009, Gourgue et al. 2009, Nikolos & Delis 2009, Marras et al. 2018) and is sufficient to keep the code stable within the scope of this study.

*R1.4: In terms of validation, it would be nice to have an empirical proof of grid convergence and test of convergence rate for the analytical cases (Cases 2.1—2.4). The authors should run simulations with successively refined grids and report L-norms and convergence rates. Tables of L-norms could be provided as an Appendix.*

We agree and have added a new section to the Appendix in which we document the grid convergence analysis of cases 2.1-2.3. We also note that grid convergence analyses for Case 2.4 and 2.5 are singled out in Section 3.2 because simulation of tsunami inundation lies at the core of our contribution. To clarify this, we have also added the following line to the end of the introduction to the model verification and validation section (lines 191-193) to help direct readers.

We refer interested readers to the Appendix for the corresponding grid convergence analysis under these relative error norms for the first three analytical cases, and we refer readers to Section 3.2 for grid convergence analysis of the tsunami modeling in the context of the ISEC benchmark as well as the Crescent City scenario.

*R1.5: I feel that the beginning of Section 3.1 discussing the benefits of TPUs for communities with no access to HPC facilities should be moved to the introduction, because it is a good motivation for the conducted research. In that context, Behrens et al. (2022) also suggested cloud computing as a possible alternative to HPC facilites. Perhaps it's interesting to the authors.*

*Behrens et al. (2022). doi: 10.3389/feart.2022.762768*

We wholeheartedly agree with this comment; the availability of rapid numerical simulation in communities that do no have access to HPC facilities is a critical motivation for the conducted research, and deserves a note in the introduction. With that in mind, we have added a brief statement to our introduction when we speak to our reasoning for choosing TPUs.

We intentionally use a hardware infrastructure that is relatively easy to use and accessible without specific training in high-performance computing. For the TPU infrastructure that we use here, comprehensive tutorials using Google Colab are available at `https://cloud.google.com/tpu/docs/colabs`. The TPU may increasingly become a standard hardware on which physics-based machine-learning algorithms will be built (Rasp et al. 2018, Mao et al. 2020, Wessels et al. 2020, Fauzi & Mizutani 2020, Liu et al. 2021, Kamiya et al. 2022). Through its relative ease of access and potential for rapid simulation capabilities, cloud computing provides a valuable alternative to higher performance computing clusters (Behrens et al. 2022, Zhang et al. 2010), particularly for communities with limited access to local clusters.

*R1.6: Can the authors comment on the process of getting access to Google's TPUs? From the website, the cloud service seems to be a paid service. Is it similar to renting time on an AWS or Microsoft Azure?*

A valuable and needed clarification. We have added some more detailed information clarifying Cloud TPU access in our brief section about running our code with TPU access in the Appendix. Google's Cloud TPUs, with up to 8 cores, are currently available to many regions for free via Google Colab (see `https://cloud.google.com/tpu/docs/regions-zones`). The only paid component comes from the connection of the TPU to long-term storage, a necessity for providing DEM files to the TPU and managing TPU outputs. The costs for this storage are minimal but vary based on the size of simulation and the region of access.

Due to the restrictions of the TPU using Google Cloud Storage, a user must leverage Google's buckets to run the notebooks. At the time of writing this article, 8 TPU cores are readily available on Google Colab for free, but Google Cloud Storage buckets are a paid subscription service. With a computing project setup on Google Cloud and a corresponding bucket with open permissions (with steps specified in `https://cloud.google.com/storage/docs/creating-buckets`), users can quickly run any of the example notebooks or design their own simulation. Any of the example notebooks available on GitHub (with the exclusion of `tpu_tsunami.ipynb`, which contains the full implementation with all of the different scenarios; and `Create_Scenarios.ipynb`, which can aid users in generating a custom DEM file) can be quickly ran by going through the notebook after a few setup steps.

*R1.7: In section 3.3, the authors should briefly report the formal accuracy of GeoClaw.*

We appreciate the reviewer's suggestion for noting GeoClaw's own accuracy, and agree that some accuracy information should be shared. We have clarified the accuracy, time-stepping, and computational resources of our GeoClaw runs, and have included a brief grid convergence analysis.

For comparison purposes, we run GeoClaw (Clawpack Development Team 2020, Mandli et al. 2016, Berger et al. 2011) using 4 CPU threads that we were allocated for free via Google Colab. Our GeoClaw simulation uses the same DEM file and is computed at an 8 m resolution without mesh-refinement. We use an adaptive time-step bounded by the Courant–Friedrichs–Lewy condition, and determine the spatial fluxes using a second-order, rate-limited Lax-Wendroff scheme. In Fig. 14, we see lower relative error norms as we approach higher resolution in our simulation of our Crescent City scenario. The GeoClaw numerical solution can be compared to our TPU numerical solution in Fig. 13, where the top row includes several instances in time of the TPU numerical solution, and the bottom row depicts the GeoClaw numerical solution at the same

instances in time. Although some differences can be seen in inundation by $t = 380\ s$ in the rightmost plots, the solutions do generally appear similar over time, lending credibility to the validity of the numerical solution presented in this paper.

[Figure]

Figure 13: TPU solution (top row) at several time instances compared to the GeoClaw solution (bottom row). The arrival of the tsunami front ($t = 100, 180$ s), the inundation of the harbor ($t = 260$ s), and coastal inundation and reflection is depicted, and relatively comparable.

[Figure]

Figure 14: GeoClaw relative error norms for a 100 s simulation for our Crescent City Scenario under varying resolutions. 4 m resolution is used as the benchmark.

*R1.8: I suggest that some part of the discussion could be separated as conclusions. I think the part starting with "Though just a starting point ..." on about line 359 on page 22 marks the end of discussion of results and starts the conclusions and outlook. But the authors may disagree.*

We agree that the last several paragraphs of the discussion reads like a conclusion. We have followed the suggestion, and separated out some of the content into a conclusions section, and also removed some content that felt repetitive in the discussion. What follows is the content of the brief Conclusion section.

**Conclusion**

We present a first step towards an accessible software package that leverages the powers of Cloud-based TPU computing for improving the capabilities of risk managers and communities to mitigate the destructive onshore impacts of tsunamis. We verify and validate our current implementation to ensure that it is capable of simulating inundation from a Carrier N-wave over real topography. These simulations are comparable to that ran by the popular open-source solver GeoClaw (Clawpack Development Team 2020, Berger et al. 2011), but can be run at higher speeds through Google Colab and requires less expertise in scientific computing. As a result, high quality tsunami simulations are available to remote communities for rapidly evaluating different risk-mitigation options including but not limited to nature-based solutions. Future efforts can then be dedicated to better meeting the needs of risk managers with a platform available through the cloud, be that in coupling our shallow water equations package to earthquake-tsunami generation models, or experimenting with different numerical implementations to enable even more rapid simulation of these equations.

**Response to Reviewer #2**

*R2.Section 3.1: In the section "3.1 Number of TPU cores", the scalability of the model is analyzed by running the Crescent City case with a different number of TPU cores (strong scaling, see https://hpc-wiki.info/hpc/Scaling). However this should be further elaborated. For sake of clarity, I suggest adding a row in Table 1 showing the speed up versus using a single TPU and then analyze the results observed. Also, a plot showing how the measured speed-up compares with an ideal scaling could be added.*

*There was no indication of the specific hardware employed. As far as I know there are several generations of TPU available with different capabilities (see https://ieeexplore.ieee.org/document/9499913). The reader may need this information in order to reproduce the experiments or to know what performance should be expected when using the model for their own application.*

*It is also unclear how the time was measured, the manuscript states that "we first measure the wall-clock time of a simulation" and the table caption adds: "This runtime excludes transfer times between the CPU and TPU". Does this mean that the preprocessing was excluded and only the main loop of the simulation was measured? Please, clarify*

We thank the reviewer for these thoughtful suggestions to emphasize clarity and transparency in our TPU usage. Throughout the paper where specific hardware was utilized and performance is of key importance, we have specified that we were assigned a TPUv2 via Google Colab. Other users may see differences in performance and capabilities based on the TPU device they are allocated. Additionally, we have clarified what we mean by 'wall-clock time', and have moved towards measuring the wall-clock time of each time-step. The updated section 3.1 reads:

As noted in the introduction, in communities where users may not have access to high performance computing facilities, the Cloud TPU Platform provides a particularly valuable resource where users can perform large-scale computations rapidly. To quantify the potential speed-up enabled by TPUs with increasing numbers of cores, we observe the average wall-clock time taken in computation for each time-step with the exclusion of the first time-step. This first time-step includes several preprocessing functions, such as reading DEM files into TPU memory, setting up initial conditions, and initializing the Tensorflow workflow. Similarly, we calculate runtime based on the amount of time spent in computation with the exception of this first step, with time which is variable from run-to-run. As shown in Table **??**, the problem size posed by the realistic scenario is sufficient to see rapid improvements in runtime based on the number of cores. We note that our analysis may vary user-to-user, depending on the TPU version that they are allocated and the number of cores available to them. Our simulations were all conducted with a TPUv2, and we extend our analysis only up to 8 TPU cores because, at the time of writing, Google Colab only provides 8 cores for free in our region.

| Number of Cores | 1 | 2 | 4 | 8 |
|---|---|---|---|---|
| Average Runtime / Time-step [ms] | 25.8 | 16.5 | 9.6 | 6.4 |
| Speedup over 1 Core [-] | * | 1.6 | 2.7 | 4.03 |

Table 1: Average TPU Runtime per time-step (in milliseconds) with varying numbers of TPUv2 cores. The Crescent City configuration at an 8 m resolution is used, with time-steps of $\Delta t = 0.02$ s for a total of 400 simulated seconds. The domain is a grid of approximately 901 by 1992 elements; TPU cores find solutions to subdomains divided in the y-direction as suggested by Hu et al. (2022) and graphically depicted in Fig. 3

*R2.Section 3.2: In section 3.2 the convergence and runtime were measured varying the spatial resolution for the Crescent City and the ISEC benchmark cases but the measurements are not properly analyzed and discussed. It looks like there is not a clear convergence, for example In the ISEC benchmark the error with 2m is lower than 1m, also it is lower using 8m than with 4m resolution. How do you interpret these observations? It looks that the errors in these resolutions vary in similar error ranges. In my opinion, even lower resolution values should be tested (e.g. 16m 25m...) to check when the error increases significantly. The Crescent City case does not have an analytical solution, but following the observations of the ISEC benchmark I would not say that the results with 2m resolution are the best, in this case and in my opinion tests with a much coarse resolution should be also tested.*

*It would be also interesting to analyze how the runtimes scale with the number of the elements of the simulation. For example, in the Crescent City the 2m case has 16x elements than the 8m case but the run time only increases 11x. In the ISEC benchmark however, from 8m to 2m, the run time is only 1.7x higher. How do you interpret this?*

*In tables 2 and 3 an "Efficiency" row is shown, however there are no clues on the text on what these magnitudes represent or how it was calculated.*

We really appreciate these constructive comments regarding section 3.2. They helped us realize that 'simulation wall-clock time' is not a particularly helpful metric for analyzing computational speedup. Because there is so much variation in the amount of time taken in preprocessing steps and reading from storage, we have shifted our focus to average time spent in each simulation time-step. In alignment with your other notes, we have expanded upon the 'efficiency' metric as part of this analysis of how the runtimes scale with the number of elements of the simulation.

By reevaluating the ISEC benchmark, we have also discovered a consequential bug in how we were calculating error with respect to the three available spatial distributions. After making this correction, we ran additional simulations at even lower resolutions, i.e., 16m and 25m and corrected our calculation of the relative error norms in the Crescent City simulation. After implementing these changes, we see behavior that appears to more properly resembles convergence. We have updated section 3.2. accordingly:

[revised manuscript text omitted]

*R2.Section 3.3: In section 3.3, the proposed model is compared with GeoClaw, however some key information is missing like the resolution used and how many TPU cores were employed in this comparison.*

Thank you for bringing this lack of information to our attention. We have added a clarification to Section 3.3 and have added a summary figure of our accuracy analysis for GeoClaw.

For comparison purposes, we run GeoClaw (Clawpack Development Team 2020, Mandli et al. 2016, Berger et al. 2011) using 4 CPU threads that we were allocated for free via Google Colab. Our GeoClaw simulation uses the same DEM file and is computed at an 8 m resolution without mesh-refinement. We use an adaptive time-step bounded by the Courant–Friedrichs–Lewy condition, and determine the spatial fluxes using a second-order, rate-limited Lax-Wendroff scheme. In Fig. 14, we see lower relative error norms as we approach higher resolution in our simulation of our Crescent City scenario. The GeoClaw numerical solution can be compared to our TPU numerical solution in Fig. 13, where the top row includes several instances in time of the TPU numerical solution, and the bottom row depicts the GeoClaw numerical solution at the same instances in time. Although some differences can be seen in inundation by $t = 380\ s$ in the rightmost plots, the solutions do generally appear similar over time, lending credibility to the validity of the numerical solution presented in this paper.

[Figure]

Figure 13: TPU solution (top row) at several time instances compared to the GeoClaw solution (bottom row). The arrival of the tsunami front ($t = 100, 180$ s), the inundation of the harbor ($t = 260$ s), and coastal inundation and reflection is depicted, and relatively comparable.

[Figure]

Figure 14: GeoClaw relative error norms for a 100 s simulation for our Crescent City Scenario under varying resolutions. 4 m resolution is used as the benchmark.

*R2.Section 3.4: In my opinion Section 3.4 is the most controversial part of the article. I'm really happy to see this analysis in the article since this is a very important topic that usually is not covered in this kind of article. However, the energy utilization estimations were made in such a naive way that the conclusions may be completely wrong:*

*At this point we don't have any information about the generation of the TPU employed for the research. This is a key point since the energy efficiency varies with the chip architecture and integration process. The number of TPU cores for this estimation was not specified also.*

*At line 298 there is a vague statement of "2 trillion operations per second (TOPS) per Watt" without specific reference. I found a reference to this in https://cloud.google.com/tpu/docs/tpus but it refers to the "Edge TPU" series that as far as I know are a different product to the TPUs used in the Google cloud. The TOPS concept itself is vague and could mean 8-bit integer or 16-bit floating point operations since they are very useful for machine learning operations that are the main purpose for this technology.*

*The authors state that 11.9 million floating point operations per time step were used, but it is unclear how they estimated this number. If I am correct, this is nearly 7 operations per element to solve SWE that looks quite low. The floating point precision used by the authors for numerical computations is not specified, taking a look at the code seems to use FP32. I haven't found any data on the TPU throughput in FP32. In https://ieeexplore.ieee.org/document/9499913 it is stated that TPUv3 has a throughput of 123 TFLOPS in BF16 with a TDP of 450W per chip (and 2 cores per chip) so in FP32 the throughput should be lower (half?). This leads us to a power usage one order of magnitude higher than stated in the paper. However, this may be wrong since the throughput values are rarely reached in real world application so the power usage would be even higher. Another approach, would be considering that the TPUv3 has 2 cores per chip and 8 cores were used for the 8m simulation during 338 seconds running at the TDP lead us to roughly 165Wh, several orders of magnitude higher than stated in the article.*

*The CPU power usage analysis has issues as well but I will not cover it in detail. It considers that a CPU that supports 8 threads (but only has 4 cores) with a TDP of 15W will use 15/8 W when using a single thread, which is completely wrong.*

*Therefore, this section (and the corresponding paragraph in the discussion) should be rewritten or removed prior to publication.*

> We greatly appreciate this comment. This section has been challenging to conceptualize and write for the reasons mentioned by the reviewer: it is both such an important component that is often overlooked and at the same so challenging to quantify meaningfully. We agree that our initial analysis was rather naive; but we share the reviewer's opinion that this type of energy usage analysis is valuable and should be more common in the literature on scientific computing. Therefore, we do wish to continue trying to get at an order-of-magnitude estimate of the power consumption benefits of using TPUs over CPUs for a simulation. We take advantage of the same referenced document and adjust our analysis with heavy inspiration from the reviewer's comments, attempting to further redirect readers to resources that may carry out this energy profiling more completely. We hope that our new analysis is up to an acceptable standard, and would love any further suggestion regarding improving it. The updated section 3.4 now reads as follows:
>
> Estimates of energy efficiency of computing operations are becoming increasingly popular, especially in response to progressing climate change (Fuhrer et al. 2018, Fourestey et al. 2014). To get a rough approximation of the comparative efficiency of a Google Cloud TPU over a CPU, we ran the Crescent City tsunami propagation problem at 8 m resolution using GeoClaw on a Google Cloud CPU node and our code on a Google Cloud TPU node. We do not have the

information nor access to the physical devices needed to conduct rigorous energy profiles to calculate efficiency as is done by others (e.g., Ge et al. 2010), so we deliver an order-of-magnitude estimate based on the Thermal Design Power of the devices that we are allocated via Google Colab. We only compared resources that are freely available through Colab in order to compare the efficiencies of computing resources that may be accessible to all users.

Based on Table 2, each time step takes an average of about $6.4 \times 10^{-3}$ seconds on the TPUv2 that we were allocated by Google Colab, corresponding to about 0.32 seconds per simulated second under our current time-stepping regime. The TPUv2 we were assigned contains 4 chips with a Thermal Design Power of 280 W per chip (Jouppi et al. 2021), meaning that each simulated-second then has an energy cost of approximately 0.1 Wh, leading to an approximately 40 Wh energy cost for a 400 simulated-second simulation. At a price of 21 cents/kWh in the U.S. at the time of writing this article, this simulation has a monetary cost of 0.84 cents.

When we ran GeoClaw for our CPU comparison on energy utilization, we enforced a fixed time step on the GeoClaw package of equal size to that of our TPU, i.e., $\Delta t = 0.02$ s, rather than leveraging GeoClaw's adaptive time-stepping to have a fair comparison in terms of the approximate number of operations. While the specific processor we were allocated for this CPU comparison is not immediately clear, we deduced from the model name, CPU family, and model number that we were allocated the Intel Xeon E5-2650 v4 with a base frequency of 2.2 GHz and a Thermal Design Power of 105 W for 12 cores and 24 threads (Intel 2016). Of this, we were allocated 2 cores/4 threads, and we took full advantage of all threads for our GeoClaw run. Each time-step took approximately 4.2 seconds, corresponding to about 205 runtime seconds per simulated second. If we assume ideal conditions leading to perfectly proportional scaling of computational speed in increasing cores, we would expect that each simulated second would take approximately 34.2 seconds when the Intel Xeon E5-2650 v4 was used to full capacity. This means that each simulated second would have an associated energy cost on the order of 1.0 Wh. A 400 modeled-second simulation would imply a total cost of approximately 400 Wh of energy, or a monetary cost of 8.4 cents. With the corresponding TPU energy calculation in mind, our conservative estimate suggests that a CPU simulation has approximately 10 times the energy cost of running an equivalent TPU simulation under the same time-stepping conditions.

While these two simulations accomplish the same thing, they have vastly different associated performances. At times, rapid computation and simulations are necessary in the context of risk analysis, and the associated energy costs of such a performant computation is worth estimating. To address this, we push our energy estimate a touch further, providing another order-of-magnitude estimate of what a CPU simulation conducted at TPU performance would be. We extrapolate our previous assumptions further, assuming proportional scaling of computational speed with increasing CPUs, and that the Thermal Design Power applies to each CPU within a system independently. Because a simulated second of a full capacity Intel Xeon E5-2650 v4 CPU takes approximately 34.2 seconds compared to the TPUv2's 0.32 seconds, over 100 CPUs at full capacity would be needed for similar rapidity in simulation. Following similar logic as done in the previous paragraph, a CPU simulation of equivalent performance would have approximately 1000 times the energy cost of running a TPU simulation when ran under the same time-stepping conditions.

*R2.Minor Comments: Figure 1: please add a north arrow.*

*Lines 161, 163, 166, 445: The year of publication is missing in the Carson reference (2002 if I am correct).*

*Eq. 9: Please, clarify why these two metrics were chosen. Explain how differently the both measure the relative error.*

*Line 275: Missing "Fig." in: . . . depicted graphically in (Fig.) 10 and in table 2.*

*Figure 10 and 11: The colors in the figure caption (purple and blue) are not the colors present in the figure (black, turquoise and lime).*

*Figure 11: The caption only describes the lower right panel. Also the chosen time instants of the simulation were not specified.*

▌ We have made these corrections.

*R2. Minor Comments: Line 88: a properly referenced link about the usage of Google Cloud TPU is encouraged. Maybe a whitepaper, an article or some documentation.*

While we share the reviewer's feelings about proper referencing, this in particular we wished to acknowledge as a unique situation. We did not provide a properly referenced link about TPU usage since most of the user-friendly tutorials provided by Google for learning about TPUs is in the form of Colab Notebooks and interactive documentation. Rather than searching for and providing a less accessible whitepaper about TPU usage, we suggest that interested readers of all backgrounds can learn about TPU utilization with the more accessible tutorials. To improve our transparency, we provide a direct URL and explain our intention for providing the reader with the URL in the paper. The suggested line now reads:

We intentionally use a hardware infrastructure that is relatively easy to use and accessible without specific training in high-performance computing. For the TPU infrastructure we use here, there are comprehensive tutorials available using Google Colab at `https://cloud.google.com/tpu/docs/colabs`.

**Response to Reviewer #3**

*R3.1: The authors present results of a case study in Crescent City, CA. They should expand their discussion of how the results can be used by mitigation planners. For example, they could show a map which indicates the high-water mark reached by the tsunami. But that clearly is directly related to the initial wave height (boundary condition). Since earthquakes can be any magnitude, the height of a resulting tsunami can vary widely. Given that, it is not clear how such a simulation is useful to planners. It could be that the steepness of the terrain is such that as the wave height increases the corresponding high water marks start to converge. A plot showing successive high water marks would be a very informative and interesting addition. Furthermore, if the relationship between the earthquake magnitude and the wave height is well known given the tectonics offshore of Crescent City, the high water marks could be labeled with the magnitude of the earthquake.*

We thank the reviewer for the suggestion. We agree that it is important to clarify how our work, despite not being a fully complete software package, could be useful to risk managers modeling coastal flooding. While not ideal and only reflecting a first step in developing the capabilities of TPU-based simulations, we have added a figure to our section 2.5 with the Crescent City scenario, indicating different high-water marks achieved by different tsunami heights from a Carrier N-wave 1 km offshore. This figure sheds light on what regions in particular are exposed to high amplitude tsunamis, a detail that could be potentially useful as risk managers evaluate which communities are at higher risk. We have added the following paragraph and a figure to our section 2.5 on lines 280-285.

The protective benefit of the mountain range can be further seen in Fig. 10. This high-water map from a 10-min simulation of runup due to the Carrier N-wave shows the spatial variation of which locations see at least 1 m of inundation under different wave amplitudes. While more work would be necessary to leverage our tsunami software package to connect the inundation by a generated wave to the forcing generated by earthquakes, a risk manager can see how higher magnitude tsunamis may disproportionately affect certain locations over others.

[Figure]

Figure 10: 1 m inundation High-water map for Crescent City, CA, under different amplitude Carrier N-waves initialized 1 km offshore. 10 minutes of simulated time. Darker shades of blue indicate the extent reached by higher amplitude N-waves.

*R3.2: Why does the error go down if the grid size is increased? I suggest the authors either explain this phenomena or further increase the grid size until the error is clearly increasing.*

We thank the reviewer for these suggestions. This exposed a miscalculation of the $L$ relative error norms, as well as the necessity to continue increasing the grid size. We have executed this suggestion in section 3.2. Some high-level analysis is added to the section, as well as updated figures. We direct the reviewer to this section 3.2, but also repeat some of that text below.

[revised manuscript text omitted]

*R3.3:Scaling data should include relative numbers, e.g., the ratio of the 2 cores result to the 1 core result.*

We have also incorporated the suggestion to include some relative numbers, particularly in the suggested table in section 3.1 on page 17 of the revised manuscript. We have also reproduced this in the response letters, please see Table 1. What follows is the corresponding text of the section.

As noted in the introduction, in communities where users may not have access to high performance computing facilities, the Cloud TPU Platform provides a particularly valuable resource where users can perform large-scale computations rapidly. To quantify the potential speed-up

> enabled by TPUs with increasing numbers of cores, we observe the average wall-clock time taken
> in computation for each time-step with the exclusion of the first time-step. This first time-step
> includes several preprocessing functions, such as reading DEM files into TPU memory, setting
> up initial conditions, and initializing the Tensorflow workflow. Similarly, we calculate runtime
> based on the amount of time spent in computation with the exception of this first step, with
> time which is variable from run-to-run. As shown in Table 1, the problem size posed by the re-
> alistic scenario is sufficient to see rapid improvements in runtime based on the number of cores.
> We note that our analysis may vary user-to-user, depending on the TPU version that they are
> allocated and the number of cores available to them. Our simulations were all conducted with
> a TPUv2, and we extend our analysis only up to 8 TPU cores because, at the time of writing,
> Google Colab only provides 8 cores for free in our region.

| Number of Cores | 1 | 2 | 4 | 8 |
|---|---|---|---|---|
| Average Runtime / Time-step [ms] | 25.8 | 16.5 | 9.6 | 6.4 |
| Speedup over 1 Core [-] | * | 1.6 | 2.7 | 4.03 |

Table 1: Average TPU Runtime per time-step (in milliseconds) with varying numbers of TPUv2 cores. The Crescent City configuration at an 8 m resolution is used, with time-steps of $\Delta t = 0.02$ s for a total of 400 simulated seconds. The domain is a grid of approximately 901 by 1992 elements; TPU cores find solutions to subdomains divided in the y-direction as suggested by Hu et al. (2022) and graphically depicted in Fig. 3

*R3.4: "leverage the newly developed Google's Tensor Processing Unit (TPU)" should be "leverage Google's newly developed Tensor Processing Unit (TPU)", although "newly" may be inapt, given the first version was developed 8 years ago.*

*Please check references, for example, line 38 should have (Gordon, 2012).*

*Earthquake should not be hyphenated (line 95).*

*Typo: line 228 has a typo "t =, "*

*Typo: line 242 "that that"*

*The authors should explain what is $\Delta\Omega$ in Eq. 9.*

*The number of simulated seconds should be mentioned re: Table 1.*

*Check table and figure references, e.g. line 275 should have "Figure 10" not "10"; line 281 "Table 3"*

*The authors tables have too many digits. It is irrelevant to have so many digits. In most cases a few is appropriate.*

*Fig. 10: line colors referenced are incorrect.*

*The authors should use scientific notation in some cases and not use it in other cases. More than 1 leading zeros: yes. No leading zeros with small magnitude number: no. Examples: 0.000749 → 7.5E-04, $4.76 \times 10^2$ → 476*

*Project Safe Havens should be Project Safe Haven*

*In the discussion (more than once) "tsunami simulation" should be "a tsunami simulation" or "tsunami simulations"*

*In their discussion of efficiency as relates to climate change, the chip efficiency is not the only concern. The authors may want to mention that Google Cloud purchases enough renewable energy to cover their entire operations. https://cloud.google.com/blog/topics/inside-google-cloud/announcing-round-the-clock-clean-energy-for-cloud*

We thank the reviewer for pointing out these errors in detail, and making so many helpful suggestions. We have corrected them in the new manuscript.